# *In-vitro* diagnostic point-of-care tests in paediatric ambulatory care: A systematic review and meta-analysis

**Oliver Van Hecke**[1]*, **Meriel Raymond**[1], **Joseph J. Lee**[1], **Philip Turner**[1], **Clare R. Goyder**[1], **Jan Y. Verbakel**[2], **Ann Van den Bruel**[2], **Gail Hayward**[1]

1 Nuffield Department of Primary Health Care Sciences, University of Oxford, Oxford, United Kingdom,
2 Academic Center for General Practice, KU Leuven, Leuven, Belgium

* oliver.vanhecke@phc.ox.ac.uk

**Data Availability Statement:** All relevant data are within the manuscript and its Supporting Information files.

## Abstract

### Introduction

Paediatric consultations form a significant proportion of all consultations in ambulatory care. Point-of-care tests (POCTs) may offer a potential solution to improve clinical outcomes for children by reducing diagnostic uncertainty in acute illness, and streamlining management of chronic diseases. However, their clinical impact in paediatric ambulatory care is unknown. We aimed to describe the clinical impact of all *in-vitro* diagnostic POCTs on patient outcomes and healthcare processes in paediatric ambulatory care.

### Methods

We searched MEDLINE, EMBASE, Pubmed, Cochrane Central Register of Controlled Trials, Cochrane Database of Systematic Reviews, and Web of Science from inception to 29 January 2020 without language restrictions. We included studies of children presenting to ambulatory care settings (general practice, hospital outpatient clinics, or emergency departments, walk-in centres, registered drug shops delivering healthcare) where in-vitro diagnostic POCTs were compared to usual care. We included all quantitative clinical outcome data across all conditions or infection syndromes reporting on the impact of POCTs on clinical care and healthcare processes. Where feasible, we calculated risk ratios (RR) with 95% confidence intervals (CI) by performing meta-analysis using random effects models.

### Results

We included 35 studies. Data relating to at least one outcome were available for 89,439 children of whom 45,283 had a POCT across six conditions or infection syndromes: malaria (n = 14); non-specific acute fever 'illness' (n = 7); sore throat (n = 5); acute respiratory tract infections (n = 5); HIV (n = 3); and diabetes (n = 1). Outcomes centred around decision-making such as prescription of medications or hospital referral. Pooled estimates showed that malarial-POCTs (*Plasmodium falciparum*) better targeted antimalarial treatment by reducing over-treatment by a third compared to usual care (RR 0.67; 95% CI [0.58 to 0.77],

**Funding:** CRG is supported by a Wellcome Trust Doctoral Research Fellowship. PT and GH are supported through the NIHR Community Healthcare MedTech and IVD Co-operative Oxford at Oxford Health Foundation Trust (award MIC-2016–018). This research was funded by the National Instituate for Health Research (NIHR) Community Healthcare MedTech and In Vitro Diagnostics Co-operative at Oxford Health NHS Foundation Trust. The funders did not have any role in the study design, data collection and analysis or preparation of the manuscript. The views expressed are those of the author(s) and not necessarily those of the NHS, the NIHR or the Department of Health and Social Care.

**Competing interests:** The authors have declared that no competing interests exist.

n = 36,949). HIV-POCTs improved initiating earlier antiretroviral therapy compared to usual care (RR, 3.11; 95% CI [1.55 to 6.25], n = 912). Across the other four conditions, there was limited evidence for the benefit of POCTs in paediatric ambulatory care except for acute respiratory tract infections (RTI) in low-and-middle-income countries (LMICs), where POCT C-Reactive Protein (CRP) may reduce immediate antibiotic prescribing by a third (risk difference, -0.29 [-0.47, -0.11], n = 2,747). This difference was shown in randomised controlled trials in LMICs which included guidance on interpretation of POCT-CRP, specific training or employed a diagnostic algorithm prior to POC testing.

## Conclusion

Overall, there is a paucity of evidence for the use of POCTs in paediatric ambulatory care. POCTs help to target prescribing for children with malaria and HIV. There is emerging evidence that POCT-CRP may better target antibiotic prescribing for children with acute RTIs in LMIC, but not in high-income countries. Research is urgently needed to understand where POCTs are likely to improve clinical outcomes in paediatric settings worldwide.

## Introduction

Point-of-care tests (POCTs) promise to revolutionise the amount and quality of care that we can deliver in the community [1]. There has been laudable progress in developing tests that are fast and simple enough to support clinical decision-making [2, 3]. These fall into two main areas: acute presentations in which a decision needs to be taken within the time frame of the consultation; and monitoring of chronic conditions, allowing advice and medication adjustments to made without the need for additional healthcare contacts. Research on the benefits of POCT has focussed on improvement of care and clinical pathways for adults [2, 4–9]. These include POCTs for cardiovascular diseases (cholesterol, NT-pro-BNP), diabetes mellitus (HbA1c and glucose), kidney disease (microalbuminuria), blood coagulation (INR and D-dimers for deep vein thrombosis and pulmonary embolism), myocardial damage (heart-type fatty acid binding protein (H-FABP), troponin, CK-MB). Here, for example, immediate POCT results are associated with the same or better medication adherence in adults compared with a laboratory-based test result [10]. However, in high-income countries (HICs), a substantial proportion (25%) of consultations in ambulatory care are for children and in particular, children with acute illness [11–13]. In low and middle-income countries (LMICs), the proportion of consultations for children with acute illness is likely to be at least this high, compounded by the disproportionate burden and mortality of infectious disease dominated by malaria, tuberculosis and HIV.

Both HIC and LMIC settings pose diagnostic challenges. The diagnostic process of acute illness and monitoring of chronic disease in children is mostly based on clinical assessment. Globally, very few children will have a serious condition requiring urgent care [14, 15], but the non-specific nature of early symptoms makes it difficult to detect those children who will progress to more serious infections and require secondary care management. This diagnostic uncertainty often leads to inappropriate prescribing, unnecessary referrals to hospital, needless additional testing [16], and a 10–20% trend increase in potentially avoidable, short stay hospital admissions of children since 1997–2012 [17–19].

In LMICs, these factors exist alongside the risk of serious communicable diseases and high childhood mortality rates. Population-level interventions, for example, like the mass roll-out of antibiotics may reduce mortality, but remains controversial and is likely unsustainable in resource-poor settings [20–22]. One factor which could help reduce this diagnostic uncertainty is POC technology. POCTs may help to improve diagnostic precision, optimise prescribing and improve the quality of care for children, and, indirectly, relieve pressure on healthcare systems [17–19]. Likewise, the benefit of POCTs for long-term conditions shorten the feedback loop by providing an immediate result that allows timely adaptation of treatment [10]. This mitigates against the impact of tardy laboratory results, or results only being actioned at the next consultation. Therefore, when treatment decisions lag behind "real-time", they often become empirical.

Yet, we should be mindful that the complexity of clinical decision-making in children is not the same as in adults. Although the analytical and clinical diagnostic accuracy of POCTs may be broadly similar in adults and children, the clinical effectiveness of using a POCT within a clinical pathway cannot be generalised from adults to children. The clinical needs are distinct. Factors such as parental concern and the potential for rapid clinical deterioration may alter the test's clinical effectiveness and diagnostic value in paediatric populations.

Currently, we do not know the existing evidence base for in-vitro POCTs in children and the clinical impact of this technology on patient outcomes and healthcare processes in paediatric ambulatory care. We therefore performed a systematic review to describe this.

## Methods

### Search strategy and inclusion criteria

The study protocol was published prospectively [23]. This review is a sub-study and evaluates the clinical impact of any in-vitro diagnostic point-of-care test (POCT) in paediatric populations in ambulatory care.

We systematically searched the six main electronic databases (MEDLINE, EMBASE, Pubmed, Cochrane Central Register of Controlled Trials, Cochrane Database of Systematic Reviews, and Web of Science) from database inception to 29 January 2020. With the help of an Information specialist, we used validated search filters for "primary care/ambulatory care", "point of care/rapid test", and "adolescent/child/infant" (example of search strategy S8 Appendix in S1 File).

We included randomised controlled trials (RCTs) and non-randomised studies of children presenting first to ambulatory care settings (general practice, hospital outpatient clinics, or emergency departments, walk-in centres, registered drug shops) where healthcare is delivered and/or POCTs are used. Children were defined by the authors of included studies.

Where studies involved both adults and children, we included studies where we were able to distinguish outcomes of children from adults. We only included studies that examined in-vitro POCTs that were defined as in-vitro i.e. tests involving blood or other bodily fluid or excreta that have been taken from the human body. Diagnostic POCTs that were not in-vitro (e.g. POC ultrasound) were excluded. Studies were eligible if they compared the POCT with usual care. Usual care could include no testing or central laboratory tests, but not another novel test or diagnostic strategy. We included and distinguished studies where POCTs were used in conjunction with another training or communication strategy.

We included all quantitative clinical outcome data across all conditions or infection syndromes reporting on the impact of POCTs on clinical care and healthcare processes. Outcome data could include: patient outcomes (e.g. mortality; morbidity); decision-making/clinical

management decisions (e.g. hospital attendance/referral); medication prescribing (e.g. antibiotic prescribing); and additional diagnostic testing.

We compared studies according to similar condition as stated by authors, study design, and outcomes. Data had to be reported in sufficient detail to compare relevant outcomes between children with similar conditions/illnesses, and a POCT versus usual care.

We excluded health economic outcomes, qualitative studies, diagnostic accuracy studies, studies solely conducted in hospital inpatient settings and hospital-acquired infections. We excluded study designs that precluded comparisons between tested and non-tested participants (case studies, case series, and studies without a suitable control).

## Analyses

Two reviewers (OVH, MR) independently screened articles in duplicate at title and abstract, and full-text levels. A third reviewer (GH) resolved any disagreement. The team (MR, GH, JJL, AvB, JV, PT, CRG) extracted data on the characteristics of included studies and assessed quality of included studies based on their respective risk-of bias tool. We used the Cochrane Risk of Bias tool for RCTs [24]. This was extended to accommodate non-randomised studies by including additional parameters such as reporting of baseline characteristics; whether intervention and control groups were similar; and whether there was a detailed description of the usual care pathway [7]. OVH checked data extraction and quality assessment. We contacted corresponding authors for clarification.

We used random effects meta-analyses (where possible) to generate pooled estimates with 95% confidence intervals (CI) for the same condition or infection syndrome. Heterogeneity was assessed using the $\chi^2$ test and $I^2$ statistic. We calculated risk ratios (RR) for dichotomous outcomes and mean differences for continuous outcomes. Subgroup analyses were performed according to study design (RCTs vs non-randomised studies). We used sensitivity analyses, excluding studies to explore heterogeneity. Results were summarised narratively where data were not sufficient to perform meta-analysis. We used Covidence software [25] for citation management. Meta-analysis was performed with Revman [26] and STATA 14 SE [27].

## Results

The searches resulted in 6,860 unique records, of which 163 full-text articles were eligible for inclusion after selection on title and abstract (Fig 1). We excluded 114 studies at the full text stage. The two most common reasons for exclusion after assessing the full text were that studies were conducted in adult populations (n = 30/114) or that clinical outcomes were not reported separately for adult and paediatric populations (n = 26/114). Finally, 49 studies satisfied our selection criteria. A systematic review on the clinical impact of influenza POCTs has been published in 2019 by our research group which included 11 studies (seven RCTs, four non-randomised studies) [6]. We found three further observational influenza studies [28–30] which do not change the overall findings of the original influenza review. As a result, we excluded the 14 influenza studies leaving 35 studies for this review.

### Characteristics of included studies

We included 35 included studies (Table 1, Fig 1).

There were 24 RCTs of which 11 were cluster RCTs [31–41], one a quasi-randomised trial (term used by study authors) [42], and the remaining 12 studies were individually randomised RCTs [5, 43–53]. Two RCTs had a 90% overlap in the population [31, 32]. In Lemiengre et al., children with episodes at a high risk of serious infection were excluded from the analysis [31]. In Verbakel et al., CRP in all children was compared to CRP only in high-risk children [32].

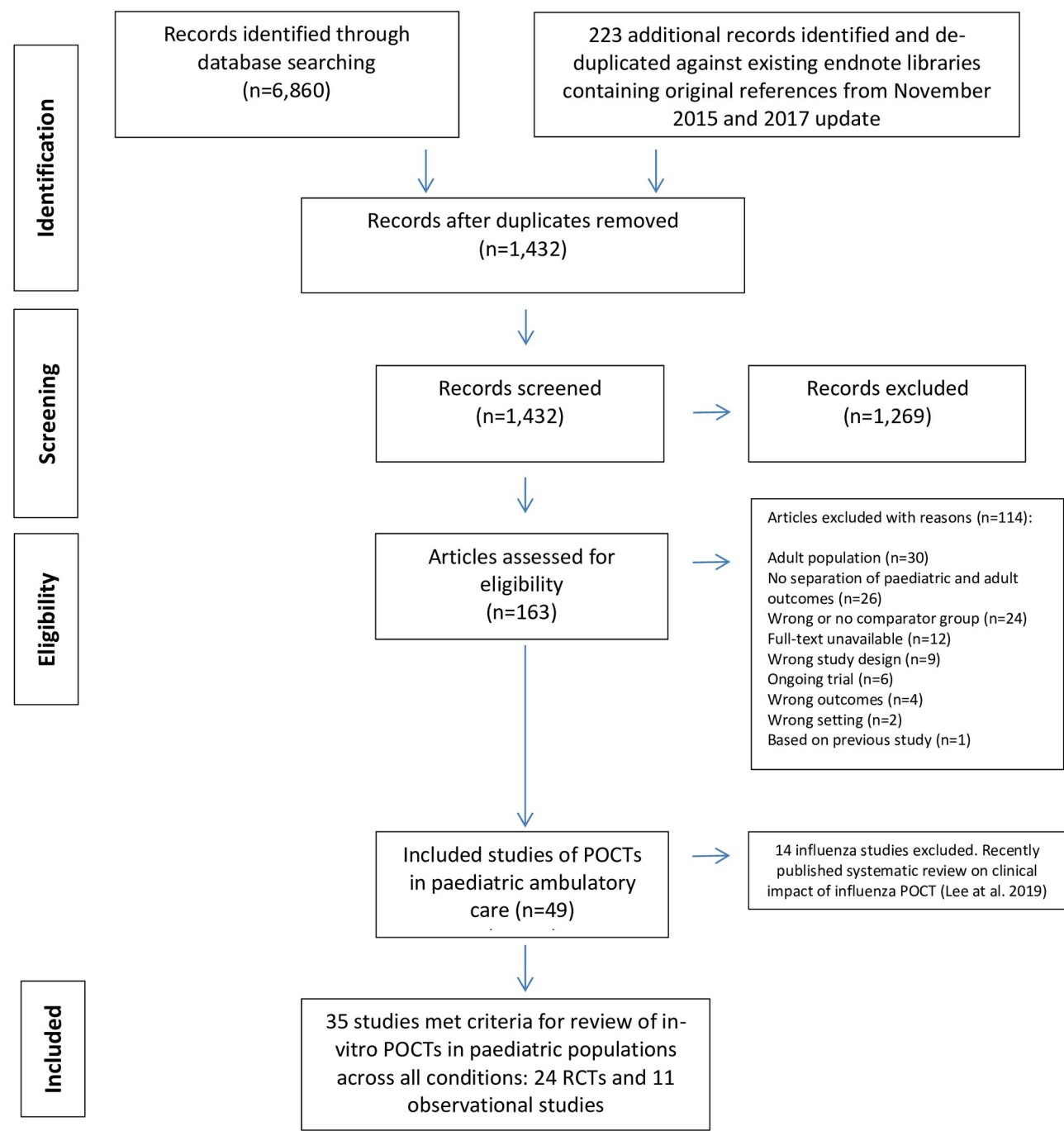

**Fig 1. PRISMA flowchart of included and excluded studies.** Abbreviations: POCTs, point-of-care tests; PRISMA, Preferred Reporting Items for Systematic Reviews and Meta-Analyses; RCT, randomised controlled trials.

We were careful to only evaluate data from either the Verbakel or Lemiengre study. There were 11 non-randomised studies. Of these, six studies compared records before and after the introduction of POCTs [54–59], three were non-randomised parallel group trials [60–62], one an observational study [63], and one a quasi-experimental study [64].

**Table 1. Characteristics of included studies.**

| Study | Design | Setting | Number of children | Point-of-care test | Role in clinical pathway | Comparator (description) | Outcomes |
|---|---|---|---|---|---|---|---|
| **Malaria (Pf)** [a] | | | | | | | |
| **Ansah et al. (2010)** | RCT | Primary healthcare clinics, Ghana | 3,957 (data from children 0–15 years old) | OptiMAL-IT rapid diagnostic test | Replacement | Two arms: (a) Microscopy (b) Clinical diagnosis | i Antimalarial treatment<br>ii Antibiotic prescribing in malaria<br>iii Safety |
| **Ansah et al. (2015)** | Cluster RCT | Registered drug shops, Ghana | 2,101 | CareStart Malaria HRP2 | Triage | Shops in communities were expected to dispense medicines without malarial POCT as per current practice | i Safety |
| **Baiden et al. (2016)** | Cluster RCT | Primary healthcare clinics, Ghana | 3,046 | CareStart Malaria First Response | Add-on | Usual care (clinical judgment) | i Mortality<br>ii Antimalarial treatment<br>iii Antibiotic prescribing |
| **Chandler et al. (2017)** | Cluster RCT | Primary healthcare clinics, Uganda | 1,336 | Not reported | Add-on | Usual care (Standard care includes services typically provided by public health centres) | i Antimalarial treatment<br>ii Antibiotic prescribing in malaria<br>iii Safety |
| **Hopkins et al. (2017)** | Observational pre- and post-implementation [Tanz-1-pub study only] [b] | Primary healthcare clinics, Tanzania | 3,454 (paediatric data from Tanz-1-pub study) | SD Bioline Pf Standard Diagnostics | Add-on/ Replacement | Period before implementation (not reported) | i Antibiotic prescribing in malaria |
| **Lal et al. (2016)** | Cluster RCT | Primary healthcare clinics, Uganda | 23,104 | First Response Malaria HRP2 | Add-on | Usual care (Presumptive diagnosis for malaria based on clinical symptoms) | i Referral |
| **Mbonye et al. (2015)** | Cluster RCT | Registered drug shops, Ghana | 8,781 | Not reported | Add-on | Usual care (Presumptive diagnosis for malaria based on clinical symptoms) | i Antimalarial treatment<br>ii Prompt antimalarial treatment within 24 hrs |
| **Msellum et al. (2009)** | Non-randomised crossover study | Primary healthcare clinics, Tanzania | 1,453 | Paracheck Pf | Add-on | Usual care (symptom-based clinical diagnosis) | i Antimalarial treatment<br>ii Antibiotic prescribing in malaria |
| **Mubi et al. (2011)** | Cross-over RCT | Primary healthcare clinics, Tanzania | 1,505 | Paracheck Pf | Add-on | Usual care (Clinical diagnosis) | i Antimalarial treatment<br>ii Referral<br>iii Mortality<br>iv Patient recovery |
| **Mukanga et al. (2012)** | Cluster RCT | Primary healthcare clinics, Burkina Faso, Ghana, Uganda | 4,216 | Multiple (First Sign Malaria Pf Card Test; Paracheck Pf; ICT Malaria Pf) | Add-on | Usual care (presumptive treatment) (presumptive diagnosis for malaria based on clinical symptoms | i Antimalarial treatment<br>ii Antibiotic prescribing in malaria<br>iii Patient recovery<br>iv Safety |

*(Continued)*

**Table 1.** (*Continued*)

| Study | Design | Setting | Number of children | Point-of-care test | Role in clinical pathway | Comparator (description) | Outcomes |
|---|---|---|---|---|---|---|---|
| **Ndyomugyenyi et al. (2016)** | Cluster RCT | Primary healthcare clinics, Uganda | 2,575 | First Response Malaria HRP2 | Add-on | Usual care (presumptive treatment) | i Antimalarial treatment<br>ii Prompt antimalarial treatment within 24 hrs<br>iii Safety |
| **Sayang et al. (2009)** | Non-randomised parallel group trial | One primary healthcare clinic, Cameroon | 312 | Diaspot Malaria RDT cassette | Add-on | Usual care (presumptive treatment) | i Antimalarial treatment<br>ii Patient recovery |
| **Ukwaja et al. (2010)** | Quasi-experimental | One primary healthcare clinic, Nigeria | 100 | Paracheck Pf | Add-on | Usual care (all children received oral antimalarial treatment in control group) | i Antimalarial treatment<br>ii Early clinic reattendance<br>iii Patient recovery |
| **Yeboah-Antwi et al. (2010)** | Cluster RCT | Primary healthcare clinics, Zambia | 3,047 | ICT Malaria Pf | Add-on | Usual care (presumptive treatment) | i Antimalarial treatment<br>ii Hospitalisation<br>iii Additional antibiotics<br>iv Mortality |
| **Non-specific fever 'illness'** | | | | | | | |
| **Althaus et al. (2019)** | 3-arm RCT | Primary healthcare clinics, and one outpatient department in Thailand, Myanmar | 1,201 | Nyocard II Reader (Axis Shield) | Triage (Two pre-defined CRP-POCT thresholds before medical examination | Usual care (described as "standard prescribing practice") | i Immediate antibiotic prescribing<br>ii Additional antibiotic prescription within 14 days |
| **Cohen et al. (2008)** | Non-randomised parallel group trial | Ambulatory paediatric private practice, France | 227 | Nyocard CRP analyser | Replacement | Usual care (laboratory CRP testing) | iii Immediate antibiotic prescribing<br>iv Hospital attendance c<br>v Additional test use |
| **Lemiengre et al. (2018)** | Cluster RCT | General practice, Belgium | 2,227 | Afinion AS 100 CRP analyser | Add-on | Usual care (not reported) | i Immediate antibiotic prescribing<br>ii Hospital attendance |
| **Nijman et al. (2015)** | Observational pre- and post-implementation | Single ED, The Netherlands | 1,939 | Afinion AS 100 CRP analyser | Triage | Period before implementation (laboratory CRP at the discretion of the ED clinician) | i Immediate antibiotic prescribing<br>ii Hospital attendance<br>iii Additional test use |
| **Rebnord et al. (2017)** | RCT | Out-of-hours general practice Norway | 397 | QuikRead Go CRP (Orion Diagnostica) | Add-on | Usual care (POC CRP at clinician's discretion) | i Immediate antibiotic prescribing<br>ii Hospital attendance |

(*Continued*)

**Table 1.** (Continued)

| Study | Design | Setting | Number of children | Point-of-care test | Role in clinical pathway | Comparator (description) | Outcomes |
|---|---|---|---|---|---|---|---|
| **Van den Bruel et al. (2016)** | RCT | Out-of-hours general practice, UK | 54 | Afinion AS 100 CRP analyser | Triage | Usual Care (usual practice) | i Immediate antibiotic prescribing |
| **Verbakel et al. (2016)** | Cluster RCT | General practices, Belgium | 3,147 | Afinion AS 100 CRP analyser | Triage | Usual Care (clinically-guided CRP testing) | i Hospital attendance |
| **Acute RTIs** | | | | | | | |
| **Diederichsen et al. (2000)** | RCT | General practices, Denmark | 139 | Nycocard CRP II (Axis Shield) | Add-on | Usual Care (clinical assessment only) | i Immediate antibiotic prescribing |
| **Do et al. (2016)** | RCT | Primary healthcare clinics, Vietnam | 1,028 | Nyocard CRP analyser | Add-on | Usual care (treated according to routine practice and local treatment guidelines) | ii Immediate antibiotic prescribing<br>iii Subsequent antibiotic at re-consultation<br>iv Change in antibiotic regime |
| **Doan et al. (2009)** | RCT | ED in tertiary hospital, Canada | 199 | Viral panel test for Adenovirus, Influenza A/B, parainfluenza 1/2/3, RSV | Triage | Usual Care (POC swab at discretion of clinician) | i Immediate antibiotic prescribing<br>ii Subsequent antibiotic at re-consultation<br>iii Length of stay in ED<br>iv Additional test use<br>v Reattendance<br>vi Ancillary tests at re-consultation |
| **Keitel et al. (2019)** | Subgroup analysis of RCT *febrile patients with non-severe respiratory symptoms | Public outpatient clinics, Tanzania | 1,726 | Two-step diagnostic algorithm (ePOCT) followed by an POCT- CRP (BioNexia CRPplus) | Add-on | Decision algorithm (ALMANACH) control arm (New Algorithm for Managing Childhood Illness Using Mobile Technology (ALMANACH) | i Immediate antibiotic prescribing<br>ii Hospital attendance<br>iii Subsequent antibiotic at re-consultation<br>iv Patient recovery<br>v Mortality |
| **Schot et al. (2018)** | RCT | General practices, The Netherlands | 309 | Afinion POC CRP (Alere Technologies AS, Oslo, Norway), | Add-on | Usual care: (GPs were advised not to use POC CRP, and treatment decisions were based on clinical assessment as usual.) | i Immediate antibiotic prescribing<br>ii Subsequent antibiotic at re-consultation within same illness period<br>iii Subsequent antibiotic at re-consultation within 3 months |
| **Acute sore throat** | | | | | | | |
| **Ayanruoh et al. (2009)** | Retrospective record review | Paediatric ED, USA | 8,280 | Rapid streptococcal test | Replacement | Period before implementation (clinical assessment only) | i Immediate antibiotic prescribing |

*(Continued)*

**Table 1.** (Continued)

| Study | Design | Setting | Number of children | Point-of-care test | Role in clinical pathway | Comparator (description) | Outcomes |
|---|---|---|---|---|---|---|---|
| **Bird et al. (2018)** | Observational pre- and post-implementation | ED in tertiary hospital, UK | 605 | Diagnostic algorithm, clinical scoring system, bionexia rapid streptococcal test | Add-on | Period before implementation (clinical assessment only) | i Immediate antibiotic prescribing |
| **Malecki et al. (2017)** | RCT | Primary healthcare clinics, Poland | 1,307 | OSOM Strep A test | | Usual care (decision to prescribe an antibiotic was based on history and physical examination) | i Immediate antibiotic prescribing ii Re-consultation |
| **Maltezou et al. (2008)** | Quasi RCT | Ambulatory paediatric private clinics, Greece | 820 | BD Link 2 Strep A Rapid antigen test | Add-on | Usual care (evaluation of children and decision to prescribe antibiotics by clinical criteria only, as in their usual everyday clinical practice) | i Immediate antibiotic prescribing |
| **Meier et al. (1990)** | Retrospective record review | Single community health centre, USA | 176 | Latex agglutination antigen detection method (Culturette, Marion Laboratories) | Replacement | Period before implementation (usual care not reported) | i Immediate antibiotic prescribing |
| **HIV** | | | | | | | |
| **Bianchi et al. (2019)** | Observational pre- and post-implementation | Cameroon, Côte d'Ivoire, Kenya, Lesotho, Mozambique, Rwanda, Swaziland, and Zimbabwe | 792 HIV positive infants (cohort of 20,865) | m-PIMA HIV-1/2 Detect (Abbott Laboratories; Lake Forest, IL, USA) or Xpert HIV-1 Qual (Cepheid; Sunnyvale, CA, USA) | Replacement | Period before implementation with conventional EID tests | i Initiating antiretroviral (ARV) therapy within 60 days |
| **Jani et al. (2018)** | Cluster RCT | Rural and urban primary healthcare clinics, Mozambique | 277 HIV positive children (cohort of 3,910) | Alere q HIV 1/2 Detect System | Replacement | Usual care (all HIV-exposed infants who presented at regular consultation visits) and existing laboratory testing | i Initiating antiretroviral (ARV) therapy within 60 days ii Retention of patients remaining on ARV |
| **Mwenda et al. (2018)** | Observational study | Ambulatory healthcare facilities, Malawi | 76 HIV positive children (cohort of 1,762) | Alere q HIV 1/2 Detect System | Replacement | Usual care and existing laboratory testing | i Initiating antiretroviral (ARV) therapy within 60 days |
| **Insulin-dependent diabetes mellitus** | | | | | | | |
| **Agus et al. (2010)** | RCT | Paediatric outpatients, USA | 215 | POCT Hba1C, DCA2000+ Analyser | Replacement | Usual care (laboratory Hba1c available several days after clinic visit) | i Change in Hba1c from baseline ii Patient communication between clinic visits |

[a] Pf: Plasmodium Falciparum;

[b] In Tanz1-pub, microscopy was available in some higher-level facilities but was not frequently use;

[c] Hospital attendance includes referral to hospital and hospital admission.

There were six conditions or infection syndromes (number of studies): malaria (n = 14); non-specific acute fever 'illness' (n = 7); sore throat (n = 5); acute respiratory tract infections (n = 5); HIV (n = 3); and diabetes (n = 1). Data relating to at least one outcome were available for 89,439 children of whom 45,283 had a POCT. Two thirds of these data (66%, 58,987/ 89,439) related to suspected malaria.

We have summarised a brief description of usual care as comparator in Table 1 (extracted from the original text). This was often not clearly defined and, in most cases, this was taken to be a clinical diagnosis with no POCT used. For example, in malarial studies set in LMICs, usual care involved children prescribed antimalarials based on clinical symptoms in outpatient clinics and cared for at home. In the usual care arm, around 90% of children were prescribed antimalarials. In non-clinical settings e.g. registered drug shops, usual care was a presumptive diagnosis of malaria. Three studies used existing laboratory-based testing as comparator as part of usual care [33, 62, 63]. In other studies, e.g. acute fever 'illness', antibiotic prescribing in usual care was around 28% and based on clinical assessment (see Fig 8) [31, 50, 51, 57, 62]. In one study, usual care was a decision algorithm modelled on a set of important paediatric signs and symptoms [52].

## Locations

Overall, 33 studies were conducted in ambulatory care settings and two malaria studies were conducted in registered drug shops where medicines are dispensed [34, 38]. Twenty studies were in LMICs in Africa focussing mainly on malaria or HIV, and the remaining 15 studies in mainly high-income European countries (Table 1).

Of the 20 studies conducted in LMICs, 18 studies were conducted in primary healthcare clinics and two studies in registered drug shops.

Of the 15 studies conducted in HICs, six studies were conducted in general practice [31, 32, 45, 48, 53, 56], four studies in emergency departments [47, 54, 57, 59], two studies in private paediatric practice [42, 62], two studies in an out-of-hours community setting [50, 51], and one study in paediatric outpatients [43].

## Principle method and target analytes of POCTs

A summary and description of each POCT and target analyte is given in S1 Appendix in S1 File. All point-of-care malaria devices used an immunochromatographic assay to either detect histidine-rich protein (HRP-2) produced by *Plasmodium falciparum*, or parasite lactate dehydrogenase (pLDH, panmalarial antigen). The three HIV studies used the same nucleic acid-based HIV POCT [33, 58, 63]. All studies using POCT-CRP employed a quantitative immunochemical assay for C-reactive protein. Five studies used a rapid antigen test for Strep A [42, 48, 54, 56, 59].

## POCTs and their intended role in clinical pathway and associated training

We examined the intended role of POCTs in the clinical pathway (S2 Appendix in S1 File) [65]. The majority of POCTs were defined as 'add-on' (n = 20) in which the new test is performed at the end of a clinical pathway to decide on appropriate treatment; nine studies used POCTs as 'replacement', in which the new test replaces an existing test, either as a faster equivalent test or to replace a non-point-of-care laboratory test, and six studies used POCTs as 'triage', in which the new test is used at the start of the clinical pathway excluding patients from further testing.

We tabulated any associated educational component in addition to POCT training offered (S2 Appendix in S1 File). Nine studies did not report any educational or training component,

five studies focussed only on POC training for clinicians, and the remaining studies included a package of POC training, clinical training and guideline update, and/or interpretation of POCT.

## Risk of bias assessment

Randomised trials were of moderate risk of bias (Fig 2); non-randomised studies had a higher risk (Fig 3). As we anticipated, none of the studies were able to blind participants and personnel to testing or test results. There were two cluster RCTs [32, 46]. that were able to conduct blind outcome assessment (one reported in two separate papers) [31, 32]. The non-randomised and before–after studies suffered from a high risk of selection, performance and detection bias and an unclear risk of reporting bias, as there was no protocol available.

We have categorised relevant outcome data according to four groups:

1. Patient outcomes (mortality; morbidity; patient recovery)

2. Decision-making/management decisions (hospital attendance/referral; early clinic reattendance/re-consultation i.e. the decision of the parent/carer to re-consult); length of stay; initiating therapy within time period; patient retention on therapy; Hba1c monitoring)

3. Prescribing (antimalarial treatment; antibiotic prescribing in malaria; initiating immediate antibiotic prescribing; subsequent antibiotic prescriptions; change in antibiotic regime)

4. Additional diagnostic testing (additional test use; ancillary tests at re-consultation)

As there were so few studies that evaluated the impact of POCT on patient outcomes, we have grouped these results together. The remaining outcomes are described per condition.

**Patient outcomes.**   Across all conditions, six studies specifically reported mortality and morbidity measures such as illness course [31, 35, 41, 44, 49, 52], of which one study for non-specific acute fever illness in Belgium reported no deaths during the study using POCT-CRP [31], and one study for acute RTIs in Tanzania, found there were two fewer deaths in the POCT-CRP arm than usual care (0/865 vs 2/854) [52]. This latter study, using a two-step intervention (diagnostic algorithm followed by POCT-CRP), also found that the difference for patient recovery within one week was clinically negligible between intervention and control arms respectively, 97.1% (840/865) vs 95.2% (813/854).

The other four studies (two RCTs, two non-randomised studies) all relating to malaria studies in LMICs, found no difference in deaths between POCT and usual care (S3 Appendix in S1 File) [35, 41, 44, 49]. There was no difference in patient recovery between malarial POCTs and usual care 3 to 7 days after treatment in four malaria studies (Fig 4) [39, 49, 61, 64]. The definition of patient recovery varied between studies, from self-reported full recovery [49], afebrile and negative blood smear [61], resolution of fever [39], or the absence of symptoms [64].

### (A) Malaria

**Decision-making.**   The proportion of children referred to the next level of care was significantly greater in children receiving a malarial-POCT than those in usual care based on two randomised studies (RR, 7.10 95% CI [2.3 to 21.92], $I^2$ = 95%) [37, 49]. One cluster RCT evaluating hospitalisation rate in children with malaria and pneumonia found no difference between POCTs 0.4% (4/1,017) and usual care 0.7% (14/2,108) [41].

**Prescribing.**   *Antimalarial treatmen*. There were 11 studies (8 RCTs and 3 non-randomised studies) evaluating the use of malarial-POCTS in endemic malaria areas [35, 36, 38–41, 44, 49, 60, 61, 64]. One RCT had two usual care arms (microscopy vs usual care; POCTs vs

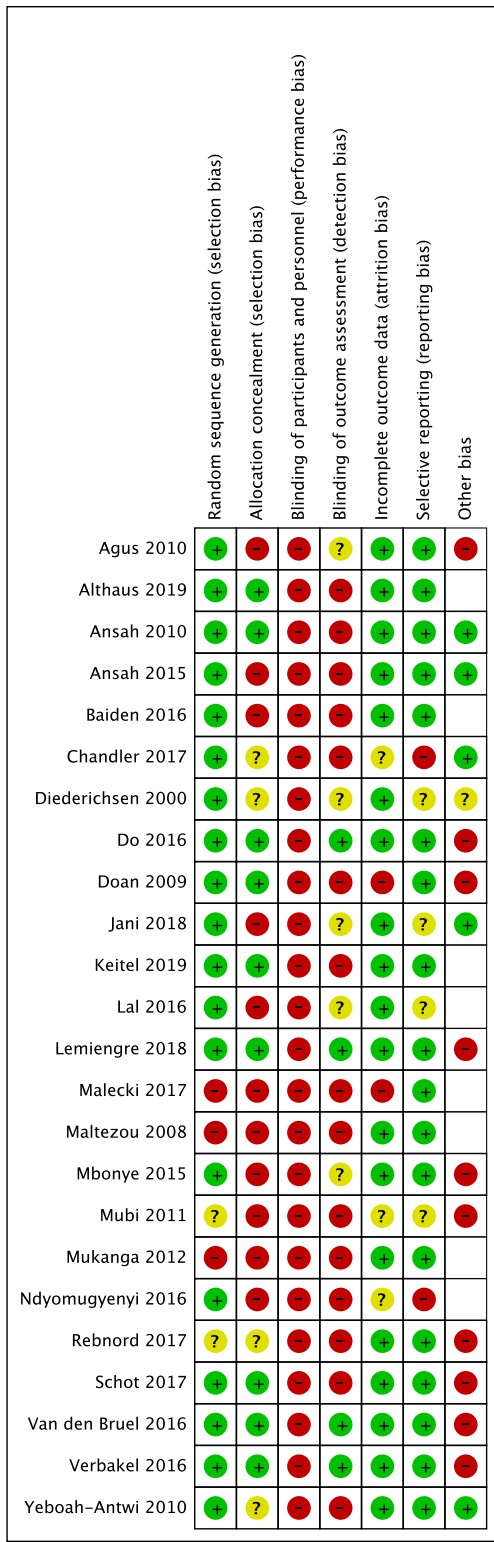

**Fig 2. Risk of bias summary for 24 randomised controlled trials across all conditions.**

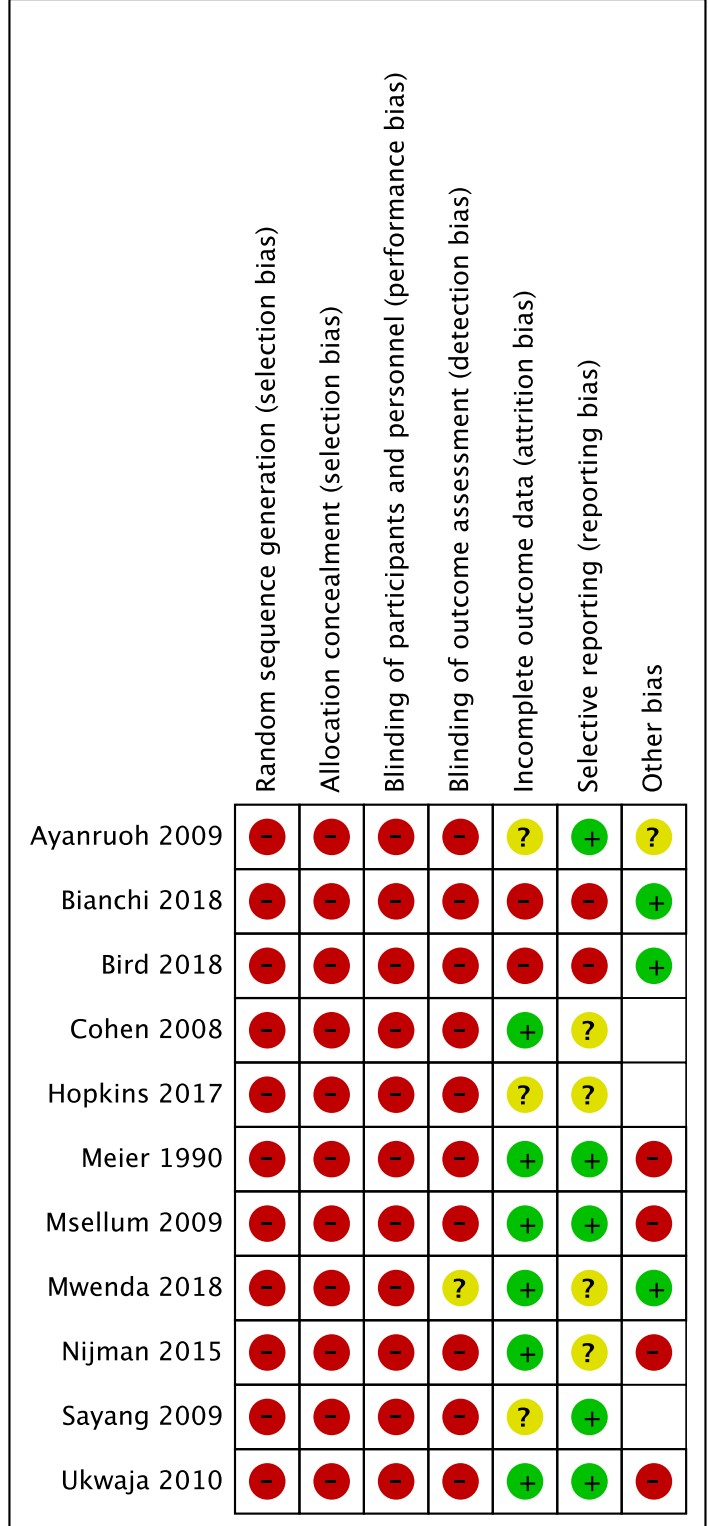

**Fig 3. Risk of bias summary for 11 non-randomised studies across all conditions.**

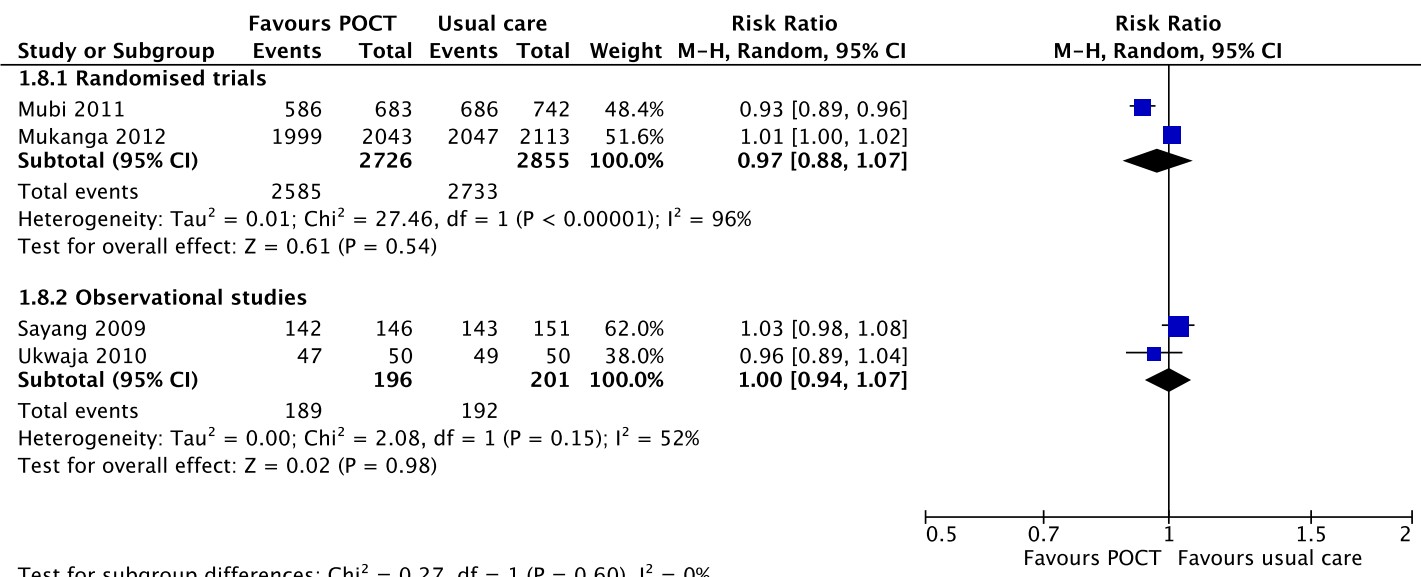

**Fig 4. Recovery between day 3 to 7.** Forest plot of meta-analyses of randomised trials and non-randomised studies reporting recovery after antimalarial treatment comparing POCT vs usual care. Abbreviations: CI, confidence interval; POCT, point-of-care test; RCT, randomised controlled trial.

usual care) in which the two comparison arms were analysed separately in the meta-analysis [44]. Pooled estimates of RCTs showed that the use of malarial-POCTs better targeted antimalarial treatment by reducing over-treatment by a third in comparison to usual care (RR, 0.67; 95% CI [0.58 to 0.77], $I^2$ = 99%) (Fig 5).

There were two RCTs which had a marked effect on the pooled estimate [40, 41]. In these two studies, almost all children in the usual care arm received antimalarials (97–99%). In addition, community health workers in the intervention arm of one of these two RCTs received additional refresher training six months after initial training [41]. Sensitivity analysis excluding these two studies showed that the effect remains significant (RR, 0.86; 95% CI [0.79 to 0.93], $I^2$ = 98%).

Based on two cluster RCTs [38, 40], the proportion of children with malaria receiving prompt and targeted antimalarial treatment within 24 hours was significantly greater with rapid diagnostic tests compared to usual care, whether that care was delivered by community health workers or registered drug shops (RR 2.72, 95% CI [1.15 to 6.43], n = 11,304, S4 Appendix in S1 File). We also summarised safety aspects of malarial-POCT interpretation and antimalarial treatment in comparison to usual care (S5 Appendix in S1 File).

*Antibiotic prescribing in suspected malaria.* There was no significant difference in antibiotic prescribing in suspected malaria cases between POCTs and usual care, based on three cluster RCTs (RR 1.04, 95% CI [0.88 to 1.22], n = 8,403, Fig 6) [35, 36, 44]. In contrast, antibiotic prescribing was more likely in usual care in two non-randomised studies [55, 60].

Based on one RCT, there was no statistically significant difference in the proportion of children who received additional antibiotics between days 5 to 7, POCTs 13/975 (1.3%) versus usual care 25/2,054 (1.2%) [41].

## (B) HIV

Three studies evaluated the use of HIV-POCT in children: one cluster RCT in Mozambique [33] and two observational studies [58, 63] set in Malawi and multiple African countries respectively.

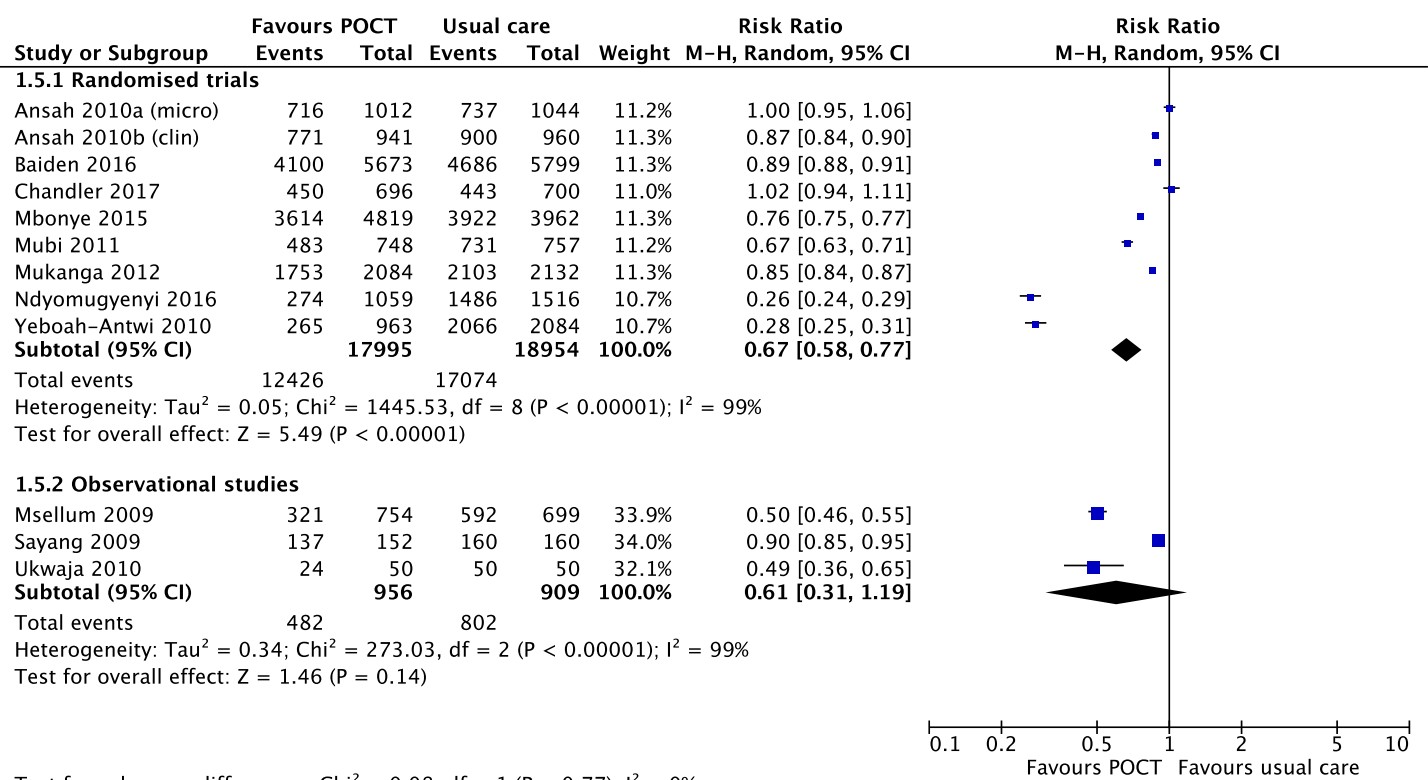

**Fig 5. Effect of malaria-POCT on antimalarial prescriptions in suspected malaria.** Forest plot of meta-analyses of randomised trials and non-randomised studies reporting a reduction in antimalarial treatment comparing POCT vs usual care. Abbreviations: CI, confidence interval; POCT, point-of-care test; RCT, randomised controlled trial.

**Decision-making.** Initiating antiretroviral (ARV) therapy within 60 days in newly-diagnosed HIV children was almost 3-fold higher in those children that had an HIV-POCT compared to usual care in three studies (RR, 3.11; 95% CI [1.55 to 6.25], p<0.001; n = 912) [33, 58, 63].

HIV-positive children who initiated ARV therapy based on HIV-POCT were also more likely to be retained in care at 90 days follow-up compared to usual care (adjusted RR, 1.40; 95% CI [1.1–1.9], p<0.027; n = 213) [33].

### (C) Non-specific acute fever 'illness'

There were seven studies that addressed non-specific acute fever 'illness' in children: one cluster RCT reported in two papers with slightly different included populations [31, 32]; three RCTs [5, 50, 51]; and two non-randomised studies [57, 66]. Studies were set in general practice (2 studies), out-of-hours setting (2 studies), primary care clinics in Thailand and Myanmar (1 study), ambulatory paediatric private practice (1 study), and the emergency department (1 study) (Table 1).

All studies used POCT-CRP (S1 Appendix in S1 File). Guidance on the interpretation of CRP results was given/available in one study [51]; intentionally not provided to clinicians in one cluster RCT (citing that safe cut-off levels in primary care are unknown) [31, 32]; subdivided into groups greater or less than CRP 60mg/L with no threshold justification [62]; subdivided into two pre-defined CRP-POCT thresholds (CRP 20mg/L and CRP 40mg/L) before medical examination [5], and not reported in two studies [50, 57]. In the one UK study,

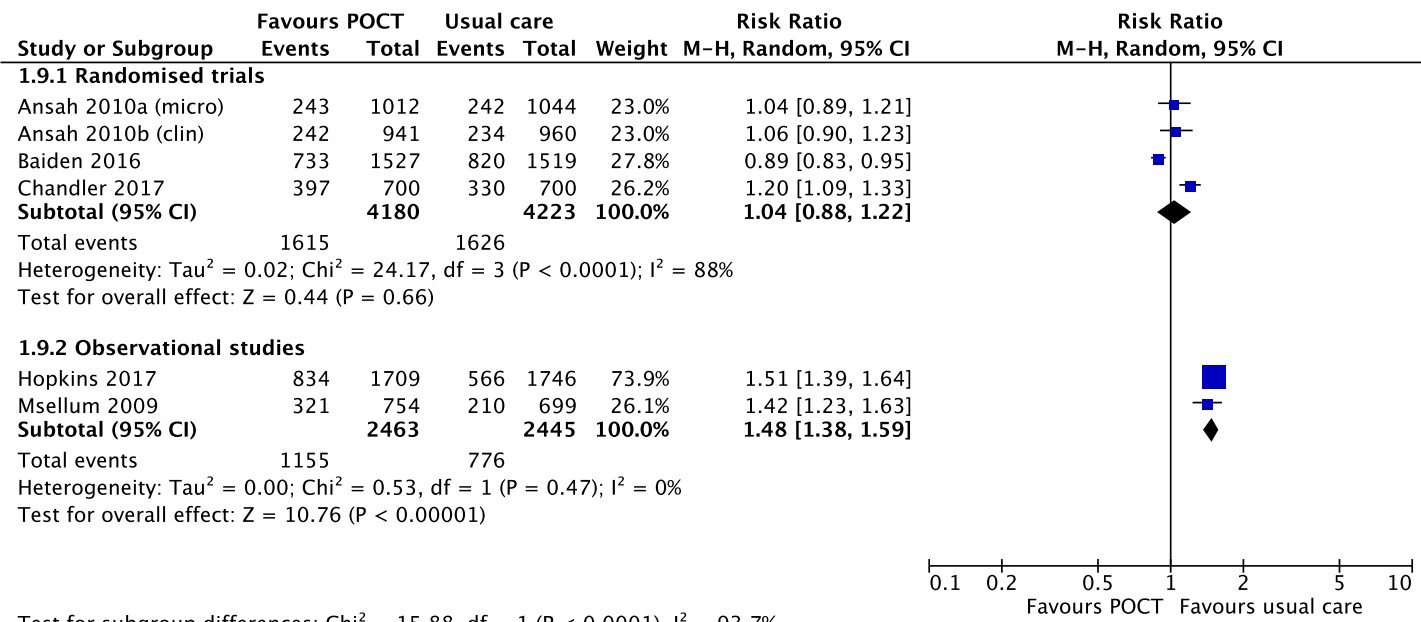

**Fig 6. Antibiotic prescribing in suspected malaria.** Forest plot of meta-analyses of randomised trials and non-randomised studies of the likelihood of antibiotic treatment in malaria comparing POCT vs usual care. Abbreviations: CI, confidence interval; POCT, point-of-care test; RCT, randomised controlled trial.

clinicians were informed that a CRP level <20 mg/L suggested a serious infection was less likely compared to a value of >80 mg/L where serious infection was more likely [51]. Additional information about CRP values can be found in S6 Appendix in S1 File.

**Decision-making.** Five studies evaluated decisions related to hospital attendance or hospital admission for non-specific acute fever illness [32, 50, 51, 57, 62]. Pooled estimates of either randomised (RR, 0.93; 95% CI [0.49 to 1.77], $I^2$ = 49%) [32, 50, 51] or non-randomised studies (RR, 0.40; 95% CI [0.12 to 1.36], $I^2$ = 81%) [57, 62] did not show a statistically significant effect on hospital attendance or admission rates (Fig 7).

**Immediate antibiotic prescribing.** Six studies reported antibiotic prescribing [5, 31, 50, 51, 57, 62]. using POCT-CRP. Neither RCTs (RR, 0.93; 95% CI [0.84 to 1.03], $I^2$ = 0%) nor non-randomised studies (OR, 0.95; 95% CI [0.83 to 1.10], $I^2$ = 0%) showed an effect of the use of POCT-CRP on antibiotic prescribing (Fig 8).

Further analysis of the Althaus study [5],where the researchers used two pre-defined POCT-CRP thresholds (CRP 20mg/L and CRP 40mg/L) before medical examination, showed that the effect on immediate prescribing was not significant for the POCT-CRP group (133/400) which used thresholds of 20mg/L (RR, 0.94; 95% CI [0.78 to 1.14]), but had an effect on immediate prescribing for the POCT-CRP group (114/399) which used thresholds of 40mg/L (RR, 0.73; 95% CI [0.54 to 0.99], p = 0.04) compared with the control group (142/402). The risk of additional antibiotic prescriptions between day 0 and day 14 in the POCT-CRP groups versus control groups was not statistically significant for either intervention group compared to the control group (CRP20 group RR, 0.94; 95% CI [0.46 to 1.92], n = 14; CRP40 group RR, 1.36; 95% CI [0.69 to 2.70], n = 20; control group n = 15).

**Test use.** Two non-randomised studies [57, 62] reported the impact of POCT-CRP on additional tests: urinalysis, blood culture, routine bloodwork, lumbar puncture and radiological imaging (Table 2).

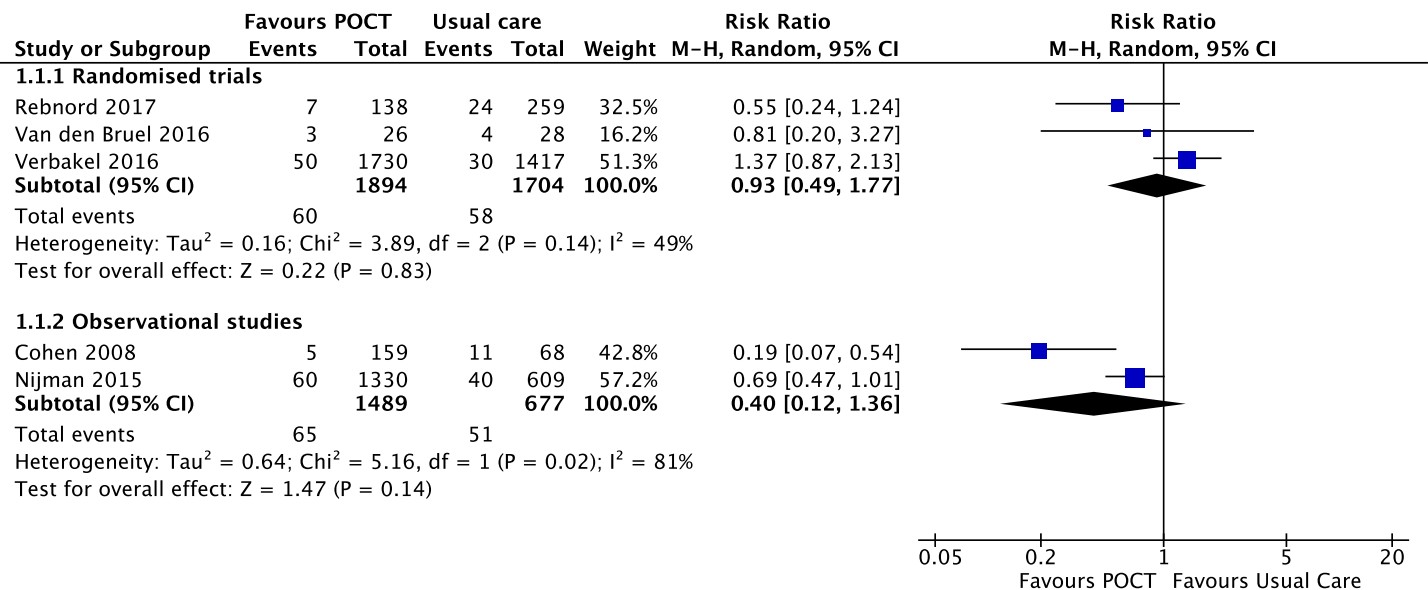

**Fig 7. POCT impact on reducing hospital attendance for non-specific acute illness.** Forest plot of meta-analyses of randomised trials and non-randomised studies reporting hospital attendance (immediate hospital assessment and/or admission) comparing POCT vs usual care. Abbreviations: CI, confidence interval; POCT, point-of-care test; RCT, randomised controlled trial.

### (D) Acute respiratory tract infections

Five RCTs focussed on acute RTIs [45–47, 52, 53]. Three studies used POCT-CRP in primary care settings in Denmark, Vietnam and The Netherlands [45, 46, 53]. One hybrid study used a two-step diagnostic algorithm (ePOCT) followed by an POCT-CRP in primary care

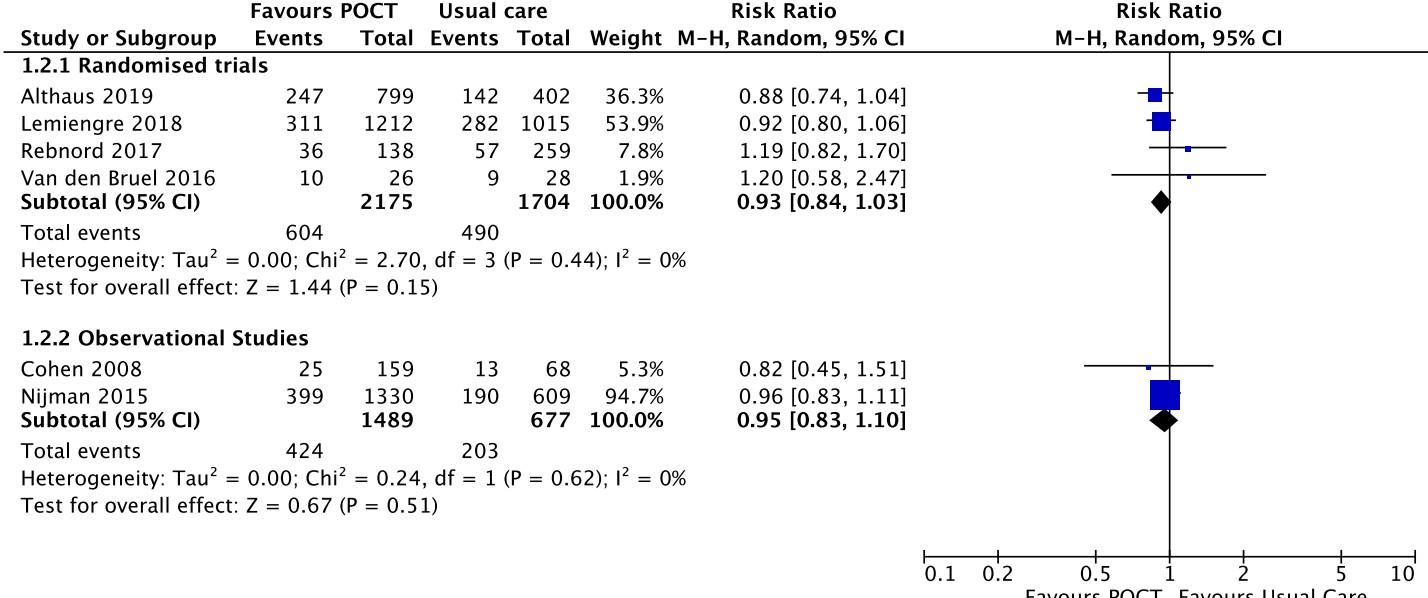

**Fig 8. Immediate antibiotic prescribing for non-specific acute fever illness.** Forest plot of meta-analyses of randomised trials and non-randomised studies reporting immediate antibiotic prescribing comparing POCT vs usual care. Abbreviations: CI, confidence interval; POCT, point-of-care test; RCT, randomised controlled trial.

**Table 2. Impact of POCTs on additional tests.**

| Outcome | Studies (n) | (Pooled) Effect estimate |
|---|---|---|
| Urinalysis | 1 (n = 227) [62] | RR, 0.29 [95% CI, 0.20 to 0. 41] |
| Blood Culture | 1 (n = 1,939) [57] | RR, 1.33 [95% CI, 0.92 to 1.94] |
| Additional blood work | 2 (n = 2,166) [57, 62] | RR, 0.21 [95% CI, 0.01 to 5.37], $I^2$ = 98% |
| Lumbar puncture | 1 (n = 1,939) [57] | RR, 0.74 [95% CI, 0.37 to 1.47] |
| Imaging (chest radiography or MRI) | 2 (n = 2,166) [57, 62] | RR, 1.25 [95% CI, 0.76 to 2.07], $I^2$ = 0% |

clinics in Tanzania [52]. One study used a viral panel POCT in a Canadian emergency department [47].

**Decision-making.**   The viral panel POCT showed no effect on re-consulting within a 7–10 day time period, and found no effect (RR, 0.88; 95% CI [0.61 to 1.27]) [47]. The duration of patient visits was not found to be different when using the viral panel POCT in the emergency department (POCT 105.7min vs usual care 156.1min; mean difference -50.4min, 95% CI [-104.6 to 3.7] [47].

The Tanzanian hybrid study (diagnostic algorithm and POCT-CRP) found that the risk of hospital admissions within thirty days was lower in the intervention arm than in the usual care arm, 0.5% (4/865) vs 1.5% (13/854), respectively (RR, 0.30; 95% CI, [0.10–0.93] [52].

**Antibiotic prescribing.**   There were four RCTs using POCT-CRP to guide immediate antibiotic prescribing for acute RTIs in children (Fig 9) [45, 46, 52, 53]. These studies were conducted in four different clinical settings (Vietnam; Denmark, Tanzania, The Netherlands) including POCT-CRP guidance and interpretation (Table 3).

In the Vietnam study [46], where CRP data for children only were available (n = 81), a third of children younger than 6 years old (n = 28) received immediate antibiotic prescription when the CRP value at enrolment was 10 mg/L or less. However, this is substantially less than in the control group of children of all ages receiving an immediate antibiotic prescription (333/518, 64·3%). In the Danish study [45], where there was no significant effect [45], there was a small sample size (n = 139), and the baseline antibiotic prescribing was almost half of that of the Vietnamese study (34%). The authors infer that the clinicians may have ignored low CRP values for prescribing antibiotics. For example, at CRP values of less than 11 mg/l, antibiotics were prescribed to 25% of patients, and at values of between 11 mg/l and 25 mg/l they were prescribed to 51% of patients. The hybrid study in Tanzania [52] children with a POCT-CRP

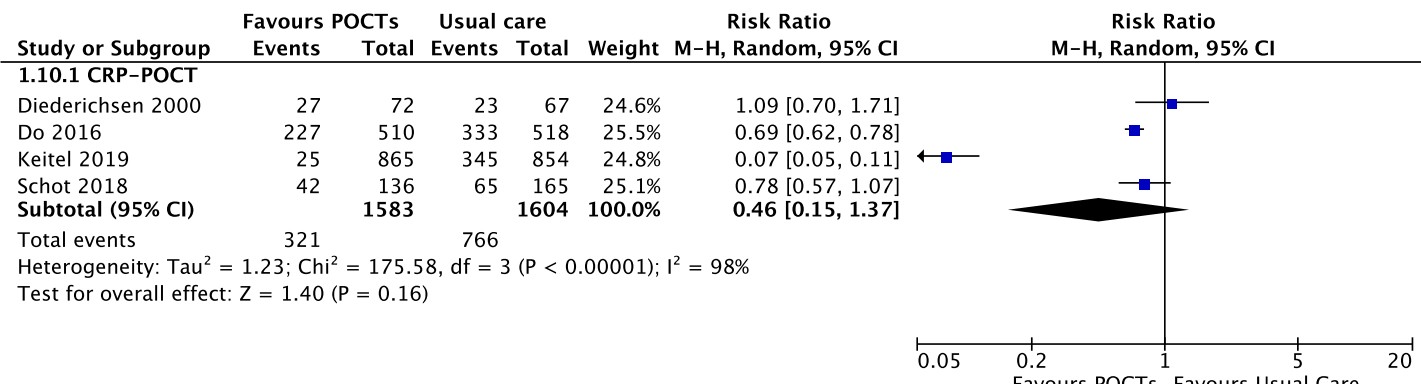

**Fig 9. Immediate antibiotic prescribing for acute respiratory tract infections.** Forest plot of meta-analyses of randomised trials and non-randomised studies reporting immediate antibiotic prescribing comparing POCT vs usual care. Abbreviations: CI, confidence interval; POCT, point-of-care test; RCT, randomised controlled trial.

**Table 3. POCT-CRP guidance and interpretation for antibiotic prescribing (acute RTIs).**

| Study | POCT-CRP guidance and interpretation for acute RTIs |
|---|---|
| Do et al. (2016) [46] | Clinicians trained to use specific CRP cut-offs: no antibiotics when the CRP level was ≤20 mg/L for patients aged ≥6 years old, and ≤ 10 mg/L for patients aged 1–5 years; referral or antibiotics when the CRP level was ≥50 mg/L. Between these thresholds no specific recommendation was given and clinicians were advised to use their clinical discretion. |
| Diederichsen et al. (2000) [45] | Clinicians informed of the normal value of CRP (<10 mg/l) and that CRP values <50 mg/l were seldom the result of bacterial infection. No strict guidelines for the use of antibiotics in relation to the CRP value were given. |
| Keitel et al. (2019) [52] | Two-step intervention, diagnostic algorithm (ePOCT) followed by POCT-CRP to inform antibiotic prescribing (combination of CRP ≥80 mg/L plus age/temperature-corrected tachypneoa and/or chest indrawing). |
| Schot et al. (2018) [53] | GPs were given the following guidance: POCT-CRP levels should be interpreted in combination with symptoms and signs; POCT-CRP levels <10mg/L make pneumonia less likely, but should not be used to exclude pneumonia when the GP finds the child ill, or when duration of symptoms is <6 hours; POCT-CRP levels >100mg/L make pneumonia much more likely, however, such levels can also be caused by viral infections; between 10mg/L and 100mg/L, the likelihood of pneumonia increases with increasing CRP levels. |

<80 mg/L were prescribed salbutamol as a home treatment in 17% (136/780) of patients in the ePOCT arm (based on a respiratory rate decrease after a salbutamol trial) and for 2% (17/769) of patients in the usual care arm. In the Dutch study [53], GPs were not provided with strict decision rules based on POCT-CRP levels, but were given guidance (Table 3). However, a relatively sample size (n = 309), protocol violations in the control group, and risk that clinicians were unblinded to the CRP level before noting a final diagnosis, which may have influenced their diagnostic labelling, limit the conclusions of this study.

There is substantial heterogeneity between the four studies described. When the Danish and Dutch studies are excluded, the findings suggest that well conducted RCTs in LMICs which include guidance on interpretation of POCT-CRP, specific training or employ a diagnostic algorithm prior to POCT-CRP testing, may reduce antibiotic prescribing by around a third (risk difference, -0.29 [-0.47, -0.11], n = 2,747) [46, 52].

There was no effect on the frequency of subsequent antibiotic prescriptions at re-consultation (day 3–5) when POCT-CRP was compared with usual care in the Vietnam study (RR, 1.16; 95% CI [0.83 to 1.61]) [46] or in the Dutch study (specified as same illness period), (RR, 0.92; 95% CI [0.33 to 2.53]) [53]. The Dutch study also assessed the effect on future consultations within the next three months, and found that 16% (13/81) of children in the POCT-CRP group consulted their GP for a new respiratory tract illness, compared to 29% (29/99) in the control group (OR 0.61; 95% CI = 0.32–1.17) [53].

In the context of the hybrid Tanzanian study, POCT-CRP underpinned by a diagnostic algorithm, led to less subsequent antibiotic prescriptions at day 7 than usual care employing another decision algorithm (RR, 0.16; 95% CI [0.12 to 0.20]) [52].

Do et al. (2016) also evaluated the effect of POCTs on subsequent antibiotic regime change but there was no statistically significant effect (RR, 1.33; 95% CI [0.41 to 4.36]) [46].

A viral panel POCT in Canada did not influence the immediate prescription of antibiotics when compared to usual care (RR, 0.86; 95% CI [0.48 to 1.53], p<0.61), but did find that fewer antibiotic prescriptions were prescribed at re-consultation within 1 week (RR, 0.36; 95% CI [0.14 to 0.95], p<0.04) [47].

**Table 4. Additional test investigations.**

| Outcome | (Pooled) Effect estimate |
|---|---|
| Urinalysis | RR, 1.12; 95% CI [0.73 to 1.71] |
| Additional blood work | RR, 0.59; 95% CI [0.28 to 1.23] |
| Imaging (chest radiography or MRI) | RR, 0.70; 95% CI [0.44 to 1.11] |
| Ancillary testing after re-consultation | RR, 0.24; 95% CI [0.03 to 1.88]** |

**Based on n = 73 children re-consulting within 7-10-day window.

**Test use.**  One study (n = 199) evaluated the effect of POCTs to detect multiple viral pathogens in acute RTIs and showed no statistically significant effect on the frequency of other test investigations (Table 4) [47]

### (E) Sore throat

Five studies focussed on POCT in paediatric sore throat: one RCT [48]; one quasi-randomised trial [42]; one pre-/post-implementation observational study [59]; and two retrospective chart review studies [54, 56]. Three studies were set in primary care and two studies in a paediatric emergency department [54, 59]. All studies used a rapid Strep A POCT.

**Decision-making.**  One RCT evaluated the effects of a Strep A test POCTs on re-consultation events [48] and found a statistically significant effect for decreasing subsequent visits when compared to usual care (RR, 4.70; 95% CI [2.94 to 7.51]; n = 1307). However, the time interval between visits was not reported.

**Antibiotic prescribing.**  Use of the Strep A POCT did not have an impact on immediate antibiotic prescribing in randomised studies (n = 2,127) [42, 48], but did show an effect in non-randomised studies (RR, 0.48; 95% CI [0.33 to 0.69], p <0.001 n = 8,717) (Fig 10) [54, 56, 59].

### (F) Diabetes Mellitus

There was one study (n = 215) involving children with insulin-dependent diabetes mellitus which evaluated POCT for glycated haemoglobin (HbA1c) on laboratory HBA1c concentrations [43].

**Decision-making.**  Over a period of 12 months, HbA1c concentrations in children with a POCT-HbA1c were initially lower when compared to the usual care group; however, at 12 months there was no significant difference. POCT-HbA1c concentrations initially decreased from baseline at 3 months (−0.20 ± 0.66%, p = 0.005) and then returned to baseline after 6 months (−0.03 ± 0.86%, p = 0.72), 9 months (+0.14 ± 0.98%, p = 0.21), and 12 months (+-0.16 ± 0.81%, p = 0.08) (S7 Appendix in S1 File). POCT-HbA1c use resulted in less frequent patient-clinician communication between visits compared to usual care (0.29 ± 0.48 vs. 0.38 ± 0.49 contacts/visit, p = 0.043).

## Discussion

### Summary of main findings

The range of conditions or illnesses for which in-vitro diagnostic POCTs have been evaluated in paediatric ambulatory care is very limited. Of the 35 studies we identified, 14 studies focused on malarial-POCTs. Only three studies focused on POCTs in other acute paediatric illness in LMICs [5, 46, 52]. Most outcomes centred around decision-making such as hospital referral or prescription of medications; mortality data and other safety data were generally not reported.

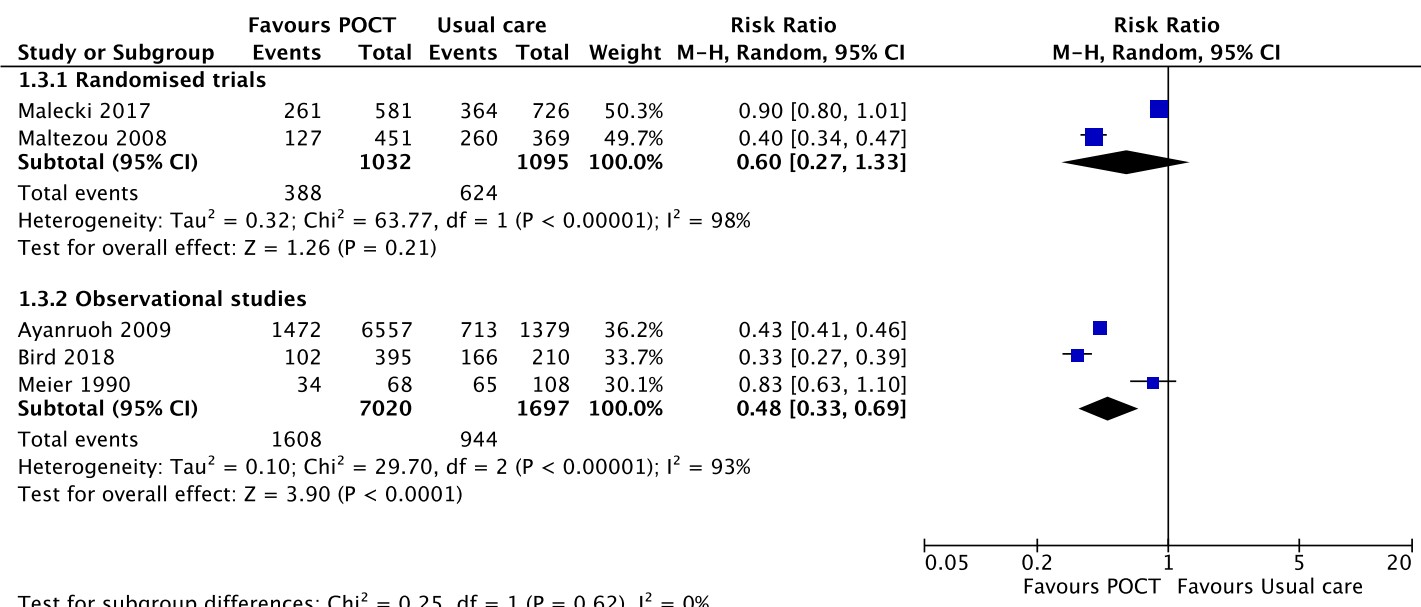

**Fig 10. Immediate antibiotic prescribing in sore throat.** Forest plot of meta-analyses of randomised trials and non-randomised studies reporting immediate antibiotic prescribing comparing POCT vs usual care. Abbreviations: CI, confidence interval; POCT, point-of-care test; RCT, randomised controlled trial.

Pooled estimates from eight RCTs showed that the use of malarial-POCTs better targeted antimalarial treatment by reducing over-treatment by a third. This is not surprising as almost all children (90%) in the usual care arm were prescribed antimalarials. However, there was no significant difference in antibiotic prescribing between children who had a malarial-POCT and those that did not for malaria cases with a suspected bacterial co-infection [35, 36, 44]. HIV-POCTs helped initiate ARV therapy early in HIV-positive children and kept them in care. This suggests that POCT can also indirectly improve access to healthcare.

POCT-CRP in undifferentiated acute fever illness did not reduce hospital attendance or admission or immediate antibiotic prescribing. Likewise, pooled estimates for Strep A POCT in sore throat did not reduce immediate antibiotic prescribing. In acute RTIs, there is some evidence that POCT-CRP may reduce immediate antibiotic prescribing in LMICs, but only in well conducted RCTs which include guidance on interpretation of POCT-CRP, specific training or employ a diagnostic algorithm prior to POC testing [46, 52].

## Interpretation of results

There are a number of factors to consider when interpreting our findings. Diagnostics are complex interventions where clinical context, patient flow, and timing affect their impact. In addition, study variability (different setting, participants, intervention design) also needs to be considered when interpreting study findings and assessing the value of a POCT.

For example, POCT-CRP in acute fever illness did not affect hospital attendance or admission. However, most studies did not offer clear guidance on the interpretation of POCT-CRP in children leaving room for variation in practice and subsequent adherence to established practice.

In acute RTI, POCT-CRP studies were conducted in very different clinical settings (Vietnam; Denmark, Tanzania, The Netherlands), using different methodology. For example, the Danish study [45], found no effect on antibiotic prescribing likely because the baseline prescription rate was so low (half that of the Vietnamese study [46]). Most outcomes did not meet

the accepted threshold of statistical significance. Many studies were underpowered to detect clinically relevant effects, or focussed on a selected population at low risk of serious infection. Other studies employed POCT-CRP in different roles in the clinical pathway in acute fever illness (e.g. triage or add-on).

These are problems that have been widely recognised as being major hurdles for diagnostic randomised controlled trials [67]. There are also likely to be important social determinants of prescribing that may override POCT-driven prescribing e.g. parental concern, the potential of rapid deterioration, and especially in LMICs, access to care [31, 68, 69]. Our data were not able to evaluate if and how parents or carers of children might influence prescribing decisions despite a 'normal' POC result.

## Comparison with existing literature

Existing literature on POCTs has focused on adult populations or mixed populations of adults and children. Our findings concur with a 2011 Cochrane meta-analysis evaluating POCTs versus clinical diagnosis of malaria in febrile mixed populations in African malaria endemic regions, where malarial-POCTs reduced antimalarial prescribing by over 50% based on four RCTs [70]. This reduction in antimalarial prescribing was more modest in our systematic review in children only (33%). One reason for this difference, might be that clinicians are more risk averse in children in LMICs where the prevalence of infectious disease, malnutrition and risk of death are greater [68]. Other reasons include perceptions that the risk of taking antimalarials is negligible for individual patients or that in high prevalence areas of malaria transmission, there is a significant false-positive malarial-POCT rate (i.e. slide negative) influencing a clinician's trust in the POCT result (due to persistent antigenaemia in individuals recently infected by malaria in hyperendemic areas [71]). Although other systematic reviews have found a reduction in antibiotic prescribing when POCT-CRP was used in adult-only and mixed populations [7, 72], we found that when POCT-CRP are evaluated in children only, there is limited evidence of benefit for their use in undifferentiated acute illness.

## Strengths and limitations

Our search strategy was comprehensive using validated search filters, and we included both RCTs and non-randomised studies conducted in ambulatory healthcare settings. We focussed on paediatric populations, an under-researched group and specifically focused on the impact of in-vitro POCTs in clinical care as opposed to diagnostic accuracy studies.

There are also important limitations. We accept that many studies showed high risk-of-bias. We had to exclude some mixed population studies where data for adults and children were inseparable and not suitable for meta-analysis. We also recognise that the distinction between acute fever illness, acute RTIs and sore throat is somewhat arbitrary, and does not necessarily reflect routine practice where infection syndromes are not always clear-cut. The data available did not allow us to sufficiently compare studies in terms of POCT-CRP thresholds. Although consultations in ambulatory care have a dual purpose—to rule out serious infections and make antibiotic prescribing decisions–the interpretation of POCT results also needs to be seen in this heterogenous context as explained above.

## Implications for clinical practice and future research

Children represent a significant proportion of consultations in ambulatory care. Yet, unlike the growing evidence in adult populations, there is a clear evidence gap for the use of POCTs to improve clinical outcomes in children worldwide. There is some evidence for POCTs in a few well-defined areas in specific settings e.g. HIV in LMICs. Yet for many other areas, mainly

in HICs, the evidence for POCTs is scarce and often at high risk of bias. Therefore, because the impact of POCTs is so context-specific, we would recommend that any implementation of POCT be closely monitored to investigate their clinical effectiveness including monitoring of any unintended consequences of testing. Failure to heed these caveats, will mean that many new tests are not routinely taken up into routine care, or are implemented despite skipping essential stages such as clinical effectiveness, and waste resources [73].

Secondly, fit-for-purpose POCTs need to be accompanied by clear guidelines on their interpretation e.g. POCT-CRP cut-off values for children. Strategies are needed to help clinicians deal with inconclusive or dubious results e.g. to aid decisions in malarial POCT-negative children who are prescribed antimalarials, or where children with acute RTIs receive antibiotics when the CRP value $\leq$ 10 mg/L. Likewise, the role of POCTs in paediatric ambulatory care will differ between LMICs and HICs in helping to guide treatment decisions in acute illness and chronic disease monitoring. For example, in LMICs, where the prevalence of serious infections is high, the role of POCTs will be to exclude serious infection. In HICs, this role may be to make antibiotic prescribing decisions based on prognosis of common (self-limiting) infections. For LMICs in particular, POCTs ought to be incorporated into existing clinical pathways e.g. the World Health Organization (WHO) Integrated Management of Childhood Illnesses (IMCI) guidelines, to ensure that there is a seamless transition [52].

Finally, this review is important to provide direction and design of future studies. Studies should expand their remit beyond malaria and HIV in LMICs and incorporate POCTs for common infection syndromes. The impact of POCTs requires careful evaluation in well-designed RCTs or other controlled study designs, taking into account that the introduction of a new diagnostic test is a complex intervention. This will require mapping the patient pathway to understand all steps from patient presentation, selection for testing, interpretation of the test result, to integration of the result in clinical decision-making. For this to be possible, qualitative and quantitative contextual information needs to be embedded in to future clinical trials. Producing studies that are too small do not guide clinicians in their interpretation and clinical decision-making, or are at high risk of bias because of methodological shortcomings, or may even lead to wrongly rejecting a valuable tool for clinical practice.

## Conclusion

There are clear evidence gaps for the use of POCTs in paediatric ambulatory care. Research has focussed on malaria- and HIV-POCTs in LMICs where they have shown benefits. There is emerging evidence that POCT-CRP may better target antibiotic prescribing for children with acute RTIs in LMICs but not in HICs. More paediatric-focussed research is urgently needed to understand where POCTs are likely to improve clinical outcomes in paediatric ambulatory settings worldwide.

## Supporting information

**S1 File.**
(DOCX)

**S1 Checklist. PRISMA-IPD Checklist of items to include when reporting a systematic review and meta-analysis of Individual Participant Data (IPD).**
(DOCX)

## Acknowledgments

The authors acknowledge Nia Roberts for her help and expertise in developing the search strategy and the reviewers for their helpful and insightful comments. The views expressed are those of the authors and not necessarily those of the National Health Service, the National Institute for Health Research (NIHR) or the UK Department of Health.

## Author Contributions

**Conceptualization:** Philip Turner, Jan Y. Verbakel, Ann Van den Bruel, Gail Hayward.

**Data curation:** Oliver Van Hecke, Meriel Raymond, Joseph J. Lee, Philip Turner, Clare R. Goyder, Jan Y. Verbakel, Ann Van den Bruel, Gail Hayward.

**Formal analysis:** Oliver Van Hecke, Meriel Raymond, Jan Y. Verbakel.

**Funding acquisition:** Philip Turner, Gail Hayward.

**Investigation:** Oliver Van Hecke.

**Methodology:** Clare R. Goyder, Ann Van den Bruel, Gail Hayward.

**Project administration:** Oliver Van Hecke.

**Supervision:** Oliver Van Hecke, Ann Van den Bruel, Gail Hayward.

**Writing – original draft:** Oliver Van Hecke.

**Writing – review & editing:** Oliver Van Hecke, Meriel Raymond, Joseph J. Lee, Philip Turner, Clare R. Goyder, Jan Y. Verbakel, Ann Van den Bruel, Gail Hayward.

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
