## [Decision Letter · Decision Letter 0]

28 Jan 2020

PONE-D-19-31785

In-vitro diagnostic point-of-care tests in paediatric ambulatory care: a systematic review and meta-analysis

PLOS ONE

Dear Dr. Oliver van Hecke,

Thank you for submitting your manuscript to PLOS ONE. Your submission has now been peer-reviewed by two experts in the field. I agree that the manuscript would benefit from being revised according to the suggestions following and encourage you to do so. 

We would appreciate receiving your revised manuscript by Mar 13 2020 11:59PM. To enhance the reproducibility of your results, we recommend that if applicable you deposit your laboratory protocols in protocols.io, where a protocol can be assigned its own identifier (DOI) such that it can be cited independently in the future. For instructions see: http://journals.plos.org/plosone/s/submission-guidelines#loc-laboratory-protocols

We look forward to receiving your revised manuscript.

Kind regards,

José Moreira, MD, MSc

Academic Editor

PLOS ONE

Journal Requirements:

2. We would strongly recommend that you update your literature search to include any relevant studies published since June 2018.

"OVH is an NIHR Academic Clinical Lecturer. JJL and MR are NIHR In Practice Fellows. CRG is

supported by a Wellcome Trust Doctoral Research Fellowship. PT and GH are supported

through the NIHR Community Healthcare MedTech and IVD Co-operative Oxford at Oxford

Health Foundation Trust (award MIC-2016–018). This research was funded by the National

Instituate for Health Research (NIHR) Community Healthcare MedTech and In Vitro

Diagnostics Co-operative at Oxford Health NHS Foundation Trust. The views expressed are

those of the author(s) and not necessarily those of the NHS, the NIHR or the Department of

Health and Social Care."

"This research was funded by the National Instituate for Health Research (NIHR) Community Healthcare MedTech and In Vitro Diagnostics Co-operative at Oxford Health NHS Foundation Trust. The funders did not have any role in the study design, data collection and analysis or preparation of the manuscript?The views expressed are those of the author(s) and not necessarily those of the NHS, the NIHR or the Department of Health and Social Care."

4. Please ensure that you refer to Figure 7 in your text as, if accepted, production will need this reference to link the reader to the figure.

5. Please upload a copy of Figure AA and BB, to which you refer in your text on page 21. If the figure is no longer to be included as part of the submission please remove all reference to it within the text.

Reviewers' comments:

Reviewer's Responses to Questions

**Comments to the Author**

1. Is the manuscript technically sound, and do the data support the conclusions?

Reviewer #1: Yes

Reviewer #2: Partly

2. Has the statistical analysis been performed appropriately and rigorously? 

Reviewer #1: I Don't Know

Reviewer #2: Yes

3. Have the authors made all data underlying the findings in their manuscript fully available?

Reviewer #1: Yes

Reviewer #2: Yes

4. Is the manuscript presented in an intelligible fashion and written in standard English?

Reviewer #1: Yes

Reviewer #2: No

5. Review Comments to the Author

Reviewer #1: This is a very well written and important manuscript. With the emergence of AMR across the world and the reduction of malaria cases in many previously highly endemic areas have put a new focus on diagnostics to guide appropriate care. Many clinical decisions are currently guided purely by clinical decision making and little is known how the right POCT at the appropriate time in the care cascade could influence outcomes and prescribing. This systematic review nicely highlights the available data but also identifies a gap in quality data, particularly for LMICs.

While this paper is well written and the analysis is presented in a very compelling way, it could benefit from some improvments:

- Abstract: While the main body of the paper is written well, the abstract needs revision. It is lacking data and clarity.

o Delete the sentence about the influenza study, not relevant in abstract

o Provide more specificity and actual data for the statements that are provided. Reduce text if needed but currently it reads like a laundry list without evidence.

o ARV: spell out in abstract

o RTIs: spell out

- Introduction:

o Rephrase the first sentence it is unusual to put so much emphasize on the In-Vitro.

o Line 88/89 and also 92: provide a reference

o Line 102: not clear why this only applies to HIC a lot of evidence suggests that this is also relevant for LMICs. See for example McDonald et al. 2018

o Line 110: why “novel” wouldn’t it be better to say “available and emerging POCTs”?

o Line 116: I don’t quite understand what the authors are saying here? Can you rephrase this for clarity? Why does treatment lag behind?

o Fig1: appears very blurry on my screen

- Results:

o What are Quasi-randomised trials? Is this a technical term? Could you define it?

o Line 229: add reference for the 28%

o Line 235: Could you provide at least the most frequent countries?

o Table 1: in the “POCT” column, could you please add the target analyte for consistency? For example: Paracheck=hrp2? QuikRead Go=?? CRP? NycoCard II… many tests exists please add CRP; latex agglutination? For what?

- Discussion:

o In the third paragraph (no line numbers…): care vs car

o 3rd paragraph in “implication” section: please add a reference or reflection on the use of clinical treatment guidelines and how POCTs should be integrated into those with clear guidance. Particularly for LMICs where IMCI guidelines exist this is relevant for uptake and use.

- Financial discloser statement includes a “?”, I suspect this is a typo?

Reviewer #2: Summary

The authors present a systematic review of studies that evaluate the impact point-of-care tests (POCT) on clinical practice, in paediatric ambulatory care. They find 30 studies, of which half evaluate the impact of malaria POCT, and other often studied conditions are non-specific fever, sore throat and acute respiratory infections. Use of malaria POCT reduced antimalarial treatment, HIV POCT increased start and retention of HIV treatment; one study in a low or middle income country (LMIC) shows a reduction of antibiotics after implementation of CRP POCT, this is not evident in high-income countries. The authors conclude there is an evidence gap for the use of POCT in ambulatory paediatric care, and that more research is needed.

The topic of evaluating impact of POCT is very relevant. A lot of research is performed to develop new diagnostic tests for infectious diseases, but it is crucial to not only show their diagnostic accuracy, but also their impact on clinical care. The authors did a good job in giving an overview on a very broad and important topic. However, I have some major concerns that have to be addressed before the study can be published.

Main concerns.

1. The literature search has been performed 1.5 years ago, which is a long time, given the fast developments in the field of POCT. Therefore, some important recent studies are missing in this review, for example studies in the impact of CRP POCT on antibiotic prescription by Keitel et al (CID 2019), Schot et al (BJGP Open, 2018) and Althaus et al (Lancet Glob Health, 2019). The last study is cited by the authors in the introduction, but not included in the review. These recent developments will influence the conclusions of the review, so the search has to be updated to be accurate and relevant.

2. The aim of the study is to review POCT and their impact on paediatric clinical care, but the results are grouped by medical conditions instead of POCTs. This reduces the clarity of the manuscript, since it is often unclear which POCT is meant. The authors already mention in the discussion that the distinction between acute illness, acute respiratory infection and sore throat is arbitrary, and in my opinion not necessary for this study. For example, CRP POCT is now covered in acute illness and in RTIs, but then also one study on a viral panel is included in RTIs. In addition, under the heading ‘malaria’ the impact of POCTs on antimalarial treatment and antibiotics is presented, which is confusing, because multiple POCTs could be used for this purpose (malaria and CRP). I suggest to restructure the manuscript and present the results by POCT instead of medical condition. In general, the manuscript often lacks precision, affecting the readability and clarity of the paper. Specific comments are provided below.

3. It is insufficiently clear in the manuscript how exactly the POCTs were used in practice in the various included studies of the review, and how the authors decided to pool the results or not. Especially the CRP studies are highly variable in how they used the test in clinical practice. Information on exact use of POCTs in the different studies needs to be added to Table 1. Then, how did the authors decide to pool these results? Did they specify criteria in terms of intended use, cut-offs or outcomes to justify the of pooling results? This is currently unclear in the methods, and not discussed in the Discussion section.

Other major issues.

Abstract

4. Please clearly state the aim in the introduction

5. Results: see main comment 2: be more specific on which POCT is meant, in line 69, 71, 73.

Introduction

6. Be more specific in what the exact evidence in adult studies is. How have POCTs impacted adult clinical care? Especially because this comes back in the conclusion ‘not show the benefits previously demonstrated in adults’. And in line 118 ‘Yet, (…) not the same as in adults’: this implies a contrast to the previous section, but the previous section does not seem to talk about adults.

7. Line 100-104: be more specific. To what does ‘both’ refer? High and low income settings? Acute and chronic diseases? ‘Few children will have a serious condition …’: in general? In acute and chronic care? This statements needs a reference.

8. Line 110-116: add references

Methods

9. Search strategy needs an update (see main comment). Moreover, please provide the exact search strategy, so that the search can be reproduced or updated by other researchers.

10. POCTs with or without training or communication strategy were not distinguished. This is a pity, since it likely makes a big difference in impact. Moreover, why did the authors then not include POCTs as part of prediction models?

11. Economic outcomes were excluded: why? Especially since the overall conclusion of the article calls for cost-effectiveness studies.

12. Hospital-acquired infections were excluded: why? Studies on inpatients were excluded already, and if children present to ambulatory care after hospitalization, the use of POCT may still be relevant.

13. Line 172 refers to another review. Is the same extended bias tool used in this study?

14. Subgroup analyses: the study protocol on PROSPERO mentions cardiac conditions, acute kidney injury, confusion, influenza, PE/DVT. This is not in line with the subgroup analysis mentioned in the manuscript. If this is because the current study is a sub-study, this needs to be explained in more detail. Also, explain why studies on influenza were published separately.

Results

15. The text mentions 6204 unique records, but figure 1 mentions 223 additional records from updates, this would be in total 6427 unique records?

16. Please add all reasons for exclusion of articles in the full text phase, also in Figure 1.

17. Refer earlier in the text to Table 1, as this is providing the overview of studies. Now the first referral to the details of POCTs is to appendix S1.

18. Line 220: influenza studies were not included at all in this review, right? Why is this mentioned explicitly here?

19. Line 223-229: description of usual care is very important in order to compare the different studies, especially since this is so variable. Please add a short description of usual care to Table 1, also if ‘not specified’ in the original study.

20. Line 228 on antibiotic prescription of 28%: to which studies does this refer?

21. line 247: ‘principle method’, unclear terminology, be more precise. Then the immunological details are not needed in the main body of the text.

22. Intended use: this is also very important when comparing POCTs. Please add a column in Table 1, describing shortly what the exact decision advice was in the included studies. Which cut-off was used? What was the advised action? Also if ‘not specified’. The current categories replacement, triage etc, do not provide enough detail.

23. Risk of bias assessment. ‘there were two cluster RCTs’; this is in contradiction with page 6 that there were 11 cluster RCTs, referring to references 23-33? At page 14 the authors refer to refs 24, 44 and 49? Then also the study of Do et al was blinded to the outcome (fig 2), so the statement that ‘one reported in two separate papers that (they?) were able to blind outcomes’ is confusing and not necessary.

24. Description of the outcome. See main comment 2: the way the results are presented is very confusing. First patient outcomes are reported (apparently defined as morbidity, mortality and recovery). Then under that heading other measures of impact were described as decision-making, prescribing and other testing. However, prescribing and doing other tests are also decision-making? To which decisions do the authors refer exactly? For malaria this is referral, for HIV it is initiating treatment, for fever hospital admission, for RTIs re-consultation? I suggest that the authors clearly specify in the Methods section which outcome measures of impact they have reviewed (it is currently not mentioned in the methods at all). It may be more clear to distinguish an impact on 1) diagnostics, 2) treatment and 3) disposition. This should be clear in Table 1 as well, as suggested in the previous comment 22.

25. Page 15 ‘in the context of safety…’ This sentence is vague, what do the authors mean hear? And why are the 3 RCTs on microscopy-positive and negative children of Table 2 not visualised in a forest plot, while their results are pooled? Other forest plots contain even fewer studies? And what is the reference standard for the effect estimate? Also after reading appendix 4 this is confusing to me. Based on the table I assume ‘microscopy results’ are the usual care? Then what is the reference standard? But in the figure of Appendix 4 the forest plot mentions POCT, when describing 3 studies on microscopy-positive children? This whole malaria treatment section needs to be revised thoroughly to improve clarity. Be consistent and specific in the terminology (POCT, usual care, rapid diagnostic tests etc)

26. Page 16 – HIV section. Confusing to mention the (absent) patient outcomes reported here again, while this topic was reported on page 14. Same holds for acute RTIs at page 18.

27. Page 17 – non-specific acute illness. See main comment 2. All studies evaluate the CRP-POCT, so I think it makes more sense to describe the CRP POCT here.

28. Page 17 – decision making. ‘(…) did not show a benefit on hospital attendance or admission’. What would be the benefit: increase or reduce? Use more neutral language, since the appropriateness of the admissions cannot be judged (like ‘there was no effect on …’). This sentence reports no effect, but then the next sentence starts with ‘However…’. This implies a contradiction, please revise.

29. Page 17 ‘We attribute the differences…’ This is for the Discussion, not results section. It does raise the question whether it makes sense at all to pool the results of such variable studies? See main comment 3.

30. Page 17 – immediate antibiotic prescribing. First sentence is unclear.

31. Page 19. The Danish study showed no effect on antibiotic prescribing, maybe because they had a lower baseline prescription rate? This is an important topic, please comment on this in the Discussion.

32. Page 19 – what do the authors mean with subsequent antibiotic prescriptions? What timeframe?

33. Page 19 ‘there was no effect on the frequency (…), however fewer antibiotics’. To what study does the second part of the sentence refer? Another than ref 44? In its current form it suggests that this was studied in the same population.

34. Page 20. How is the effect of strep A POCT on re-consultation a decision-making outcome? Whose decision? What was the advice of the POCT? In addition: how does the clinical protocol of making cultures in POCT negative children explain a rise in throat cultures? What was the usual care protocol then?

35. Page 20. Diabetes. This topic is very different from all the other POCTs that focus on infectious diseases. The authors may consider removing this topic from the review, so that it has more focus. If they decide to keep it, this sections needs to be clarified: Only the trends in HbA1c levels are provided of children who used the POCT? What was the difference with usual care? Which decision was influenced by the POCT?

Discussion

36. Summary of findings is very long, please summarize.

37. Page 21 bottom: to which figures to the authors refer? Are these new results?

38. Please include a section ‘interpretation of results’, in which the authors can discuss for example: how did the variability in studies influence the pooled results? What are reasons for finding a limited impact? It is mentioned a bit in the ‘comparison to the literature’ and ‘strengths and limitations’, but would be more clear if after the summary of findings the findings are shortly interpreted.

39. Page 22 line ‘one explanation for this difference is that…’: logic is unclear: how can a negative test result carry a high false-positive rate? And then influence the clinicians trust? (not trusting a false-positive test would then lead to less prescriptions?)

40. Page 22 ‘Another reason might be…’ this section needs a reference.

41. Page 22 ‘Strengths and limitations’ ‘Our data were not able to evaluate if and how’ This refers to the discussion of confounders in general, that might explain the lack of evidence for impact of POCT. Not only parental concern, but also clinician’s concern etc. The section where that is described (just above strengths and limitations’ is important for the interpretation of findings, and needs more references to existing literature.

42. Page 23 first line ‘dual purpose’: this is not a dual purpose, but refers to the same decision, right? By identifying serious infections you make the prescription decision.

43. Page 23 Implications. The fact that there are many children in ambulatory care does not directly justify a call for more evidence for POCTs. The fact that adult studies have proven an impact does provide more justification for that call (see also comment in introduction).

44. Page 23 ‘POCT-negative children’. Be more specific, which POCT?

45. Page 23 ‘The role of POCTs in paediatric care will differ between LMICs and HICs’. Fair point, comment on how this will be different.

Conclusion

46. The conclusion is to broad and not supported by the rest of the manuscript. It refers to benefits in adults, that has not been clearly demonstrated in the introduction or the discussion. Moreover, it calls for cost effectiveness studies, while economic outcomes were excluded from this review.

Minor issues.

- Abstract – methods: line 54 ‘compared this…’: this suggests that the authors themselves compared the POCTs to usual care, whereas they included studies that did so. Please rephrase.

- Introduction, line 96 ‘, this is likely’. To what does ‘this’ refer? The proportion of children? Acute illness?

- Line 106: 20% increase in short stay. Compared to what?

- Line 111: optimise prescribing. Of antibiotics? Please add.

- Line 206: add references of the remaining nine studies

- Line 242-244: this does not describe locations, and is already mentioned earlier, please remove.

- Line 251 POCT test, remove ‘test’.

- Table 1: there is an asterisk (*) in the row of the Nijman study that is not explained in the footnote.

- After Table 1, line numbers are missing.

- Figure 5 legend: please remove ‘inappropriate’, since this cannot be judged based on these data

- Page 16 ‘the effect of HIV-POCTs (…)’. Unclear sentence. Do the authors mean that in POCT care 4 times more children initiated ARV treatment than in usual care?

- Page 18, acute RTIs. Add reference of the Canadian ED study.

- Table 4 lacks a title.

- Page 20 – antibiotic prescribing. Be more specific. ‘Use of the Strep A POCT did not have an impact on immediate antibiotic prescribing.’

- There are no references in the manuscript to Appendices 5 and 7.

6. PLOS authors have the option to publish the peer review history of their article (what does this mean?). If published, this will include your full peer review and any attached files.

Reviewer #1: Yes: Sabine Dittrich

Reviewer #2: No

---

## [Author Response · Author response to Decision Letter 0]

6 May 2020

17 April 2020

Editor, 

PLoS One

Dear Editor 

Revision PONE-D-19-31785

In-vitro diagnostic point-of-care tests in paediatric ambulatory care: a systematic review and meta-analysis

Thank you for providing us the opportunity to improve this manuscript albeit slightly later than planned because of the current COVID pandemic. We respond to reviewer comments point by point below. Where feasible, we have incorporated the suggestions to improve the manuscript. Changes are highlighted in the revised manuscript using tracked changes.

Reviewer #1: 

This is a very well written and important manuscript. With the emergence of AMR across the world and the reduction of malaria cases in many previously highly endemic areas have put a new focus on diagnostics to guide appropriate care. Many clinical decisions are currently guided purely by clinical decision making and little is known how the right POCT at the appropriate time in the care cascade could influence outcomes and prescribing. This systematic review nicely highlights the available data but also identifies a gap in quality data, particularly for LMICs.

Thank you. 

While this paper is well written and the analysis is presented in a very compelling way, it could benefit from some improvements:

- Abstract: While the main body of the paper is written well, the abstract needs revision. It is lacking data and clarity.

We have rewritten the abstract and included specific findings which were lacking in the original submission. 

o Delete the sentence about the influenza study, not relevant in abstract 

We have deleted this sentence in the abstract. 

o Provide more specificity and actual data for the statements that are provided. Reduce text if needed but currently it reads like a laundry list without evidence.

We have added more clarity to the abstract highlighting the main findings with relevant data. 

o ARV: spell out in abstract. Done

o RTIs: spell out. Done 

- Introduction:

o Rephrase the first sentence it is unusual to put so much emphasize on the In-Vitro. 

We have deleted the word ‘in-vitro’. 

o Line 88/89 and also 92: provide a reference

We have provided references for the statements as an example of the progress that has been made with POCTs in ambulatory care. 

o Line 102: not clear why this only applies to HIC a lot of evidence suggests that this is also relevant for LMICs. See for example McDonald et al. 2018

Thank you for the useful reference. We agree that, for example, in the majority of paediatric febrile illnesses, these illnesses will be uncomplicated and self-limiting and have changed the sentence to reflect this. 

o Line 110: why “novel” wouldn’t it be better to say “available and emerging POCTs”?

Thank you. We have changed the wording as per your suggestion. 

o Line 116: I don’t quite understand what the authors are saying here? Can you rephrase this for clarity? Why does treatment lag behind?

By “treatment lagging behind”, we mean that treatment decisions are made empirically after the patient has left the consultation while waiting for laboratory test to come back. We have reworded this sentence to avoid confusion. 

o Fig1: appears very blurry on my screen

We have updated a new version of Fig 1. 

- Results:

o What are Quasi-randomised trials? Is this a technical term? Could you define it?

A quasi-randomised trial is one in which participants are allocated to different arms of the trial using a method of allocation that is not truly random. This is a known methodological term, quasi meaning ‘partly’ or ‘seemingly’. We have used this term as per the original description that the authors used to describe their study and have included this in the text. 

o Line 229: add reference for the 28%

This is based on included studies involving children with acute fever ‘illness’ (5 studies). We have added the references and signposted readers to Figure 8. 

o Line 235: Could you provide at least the most frequent countries?

Table 1 provides the reader with the most frequent high-income countries. However, we feel that adding this detailed information in the text is perhaps less informative. However, we have amended the sentence to read that the most frequent HICs were European. For information only, the number of studies conducted in HIC were: USA (3); Belgium (2); The Netherlands (2); UK (2); France (1); Denmark (1); Norway (1); Poland (1); Greece (1). 

o Table 1: in the “POCT” column, could you please add the target analyte for consistency? For example: Paracheck=hrp2? QuikRead Go=?? CRP? NycoCard II… many tests exists please add CRP; latex agglutination? For what?

The names of POCT listed in Table 1 are as described by the authors of included studies. Detailed description of POCT was often not consistent or unavailable. We have tried to homogenous these for consistency. As space is limited within Table 1, and Reviewer 2 has also suggested we include a description of ‘usual care’, we have decided to rather include a description of the target analyte for each POCT in Appendix S1. 

- Discussion:

o In the third paragraph (no line numbers…): care vs car

Thank you. We have corrected this typo. It should read ‘care’. 

o 3rd paragraph in “implication” section: please add a reference or reflection on the use of clinical treatment guidelines and how POCTs should be integrated into those with clear guidance. Particularly for LMICs where IMCI guidelines exist this is relevant for uptake and use.

Thank you for highlighting this important point. We have incorporated this and reworded the sentence. 

- Financial discloser statement includes a “?”, I suspect this is a typo? Thank you for spotting this. 

Reviewer #2: Summary

The authors present a systematic review of studies that evaluate the impact point-of-care tests (POCT) on clinical practice, in paediatric ambulatory care. They find 30 studies, of which half evaluate the impact of malaria POCT, and other often studied conditions are non-specific fever, sore throat and acute respiratory infections. Use of malaria POCT reduced antimalarial treatment, HIV POCT increased start and retention of HIV treatment; one study in a low- or middle-income country (LMIC) shows a reduction of antibiotics after implementation of CRP POCT, this is not evident in high-income countries. The authors conclude there is an evidence gap for the use of POCT in ambulatory paediatric care, and that more research is needed.

The topic of evaluating impact of POCT is very relevant. A lot of research is performed to develop new diagnostic tests for infectious diseases, but it is crucial to not only show their diagnostic accuracy, but also their impact on clinical care. The authors did a good job in giving an overview on a very broad and important topic. However, I have some major concerns that have to be addressed before the study can be published.

Thank you. This review is a bit of a behemoth and we appreciate the time you have taken to improve our manuscript. 

Main concerns.

1. The literature search has been performed 1.5 years ago, which is a long time, given the fast developments in the field of POCT. Therefore, some important recent studies are missing in this review, for example studies in the impact of CRP POCT on antibiotic prescription by Keitel et al (CID 2019), Schot et al (BJGP Open, 2018) and Althaus et al (Lancet Glob Health, 2019). The last study is cited by the authors in the introduction, but not included in the review. These recent developments will influence the conclusions of the review, so the search has to be updated to be accurate and relevant.

We agree, the field is rapidly progressing. We have therefore updated the search as planned up to 29 January 2020 and included 5 new studies: Althaus et al. 2019 (Fever); Keitel et al. 2019, Schot et al. 2018 (acute RTIs); Bird et al. 2018 (sore throat); Bianchi et al. 2019 (HIV). 

2. The aim of the study is to review POCT and their impact on paediatric clinical care, but the results are grouped by medical conditions instead of POCTs. This reduces the clarity of the manuscript, since it is often unclear which POCT is meant. The authors already mention in the discussion that the distinction between acute illness, acute respiratory infection and sore throat is arbitrary, and in my opinion not necessary for this study. For example, CRP POCT is now covered in acute illness and in RTIs, but then also one study on a viral panel is included in RTIs. In addition, under the heading ‘malaria’ the impact of POCTs on antimalarial treatment and antibiotics is presented, which is confusing, because multiple POCTs could be used for this purpose (malaria and CRP). I suggest to restructure the manuscript and present the results by POCT instead of medical condition. In general, the manuscript often lacks precision, affecting the readability and clarity of the paper. Specific comments are provided below.

We have given these suggestions some thought however restructuring the manuscript according to POCT presents its own problems. Firstly, we would prefer to keep this review clinically-focussed, and therefore structuring this according to disease/condition would be more relevant for clinicians. Secondly, as you mention in point 3 (below), even if the same POCTs are used in different studies, how these POCTs are used by health professionals also varies from study to study, and where they slot into the clinical pathway varies. Lastly, by presenting the results per POCT we assume that the tests have the same diagnostic accuracy. Therefore, presenting the results per POCT has its own limitations.

We mention the ‘arbitrary’ distinction between acute illness, acute respiratory infection etc. because paediatric illness is sometimes undifferentiated at first, however from a clinical point of view, it is sensible to describe the value of POCTs according to condition. We have taken the condition at face value as presented in the original studies and where this was described. 

We agree that more clarity is needed about which POCTs we are referring to in the Results sections e.g. malaria and CRP, and have tried to be more specific throughout the manuscript e.g. POCT-CRP. 

3. It is insufficiently clear in the manuscript how exactly the POCTs were used in practice in the various included studies of the review, and how the authors decided to pool the results or not. Especially the CRP studies are highly variable in how they used the test in clinical practice. Information on exact use of POCTs in the different studies needs to be added to Table 1. Then, how did the authors decide to pool these results? Did they specify criteria in terms of intended use, cut-offs or outcomes to justify the of pooling results? This is currently unclear in the methods, and not discussed in the Discussion section.

We agree that the studies involving POCT-CRP were highly variable and were acutely aware that simply combining these may lead to misinterpretation. We compared studies according to similar condition as stated by authors, study design, and outcomes. Data had to be reported in sufficient detail to assess relevant outcomes between patients with similar POC test and usual care. The data available did not allow us to sufficiently compare studies in terms of CRP cut-offs. The description of the exact use of POCTs in included studies was very limited or not available at all. 

However, as per your suggestion, we felt it was important to highlight the role of the POCT in the clinical pathway and have now expanded on the description of these roles (where possible) in Appendix S2.

Other major issues.

Abstract

4. Please clearly state the aim in the introduction

We have added the aim in the abstract introduction. 

5. Results: see main comment 2: be more specific on which POCT is meant, in line 69, 71, 73.

We have clarified which POCT is meant. 

Introduction

6. Be more specific in what the exact evidence in adult studies is. How have POCTs impacted adult clinical care? Especially because this comes back in the conclusion ‘not show the benefits previously demonstrated in adults’. And in line 118 ‘Yet, (…) not the same as in adults’: this implies a contrast to the previous section, but the previous section does not seem to talk about adults.

Thank you for pointing this out. We have now included some examples where POCTs have transformed clinical care in adults for comparison. 

7. Line 100-104: be more specific. To what does ‘both’ refer? High and low income settings? Acute and chronic diseases? ‘Few children will have a serious condition …’: in general? In acute and chronic care? This statements needs a reference.

We have clarified this sentence to refer to both high-income and low-income settings. We have also added references to explain the relatively low incidence of serious conditions in children. 

8. Line 110-116: add references

We have now added references to support these statements. 

Methods

9. Search strategy needs an update (see main comment). Moreover, please provide the exact search strategy, so that the search can be reproduced or updated by other researchers.

We have updated the search and included an example of the search strategy in Appendix S8 which can be modified across search platforms. 

10. POCTs with or without training or communication strategy were not distinguished. This is a pity, since it likely makes a big difference in impact. Moreover, why did the authors then not include POCTs as part of prediction models?

We have taken this suggestion onboard. We agree that most POCTs would have required some sort of training before using the POC equipment, however we are mainly interested in whether there were specific educational and communications packages that were expressly incorporated with the POCT, as opposed to the complex intervention of prediction models incorporating POCT. This is a different research question as we would be evaluating the effect of a POCT test on its own. We have reviewed all included studies again and added a detailed description of any educational and/or training packages where these data were available (Appendix S2).

11. Economic outcomes were excluded: why? Especially since the overall conclusion of the article calls for cost-effectiveness studies.

This is important to address for future studies but falls beyond the scope of the review. We focussed on patient relevant clinical outcomes. Economic outcomes of course are also relevant to patients, but more at a policy level.

12. Hospital-acquired infections were excluded: why? Studies on inpatients were excluded already, and if children present to ambulatory care after hospitalization, the use of POCT may still be relevant.

The latter point is valid but again this is a different research question and falls beyond the scope of this review. Our review focusses on the use of POCTs where children presented first in ambulatory care. 

13. Line 172 refers to another review. Is the same extended bias tool used in this study?

Yes, the same risk-of-bias tool was used. 

14. Subgroup analyses: the study protocol on PROSPERO mentions cardiac conditions, acute kidney injury, confusion, influenza, PE/DVT. This is not in line with the subgroup analysis mentioned in the manuscript. If this is because the current study is a sub-study, this needs to be explained in more detail. Also, explain why studies on influenza were published separately.

Our group has now published a number of systematic reviews assessing the impact of point-of-care tests on patients and healthcare processes. The conditions listed on Prospero are potential examples of where the group thought POCTs were use, however this review in not limited to these conditions. For sake of ease, we have subdivided these. The recently published influenza review includes adults and paediatric populations. 

 Results

15. The text mentions 6204 unique records, but figure 1 mentions 223 additional records from updates, this would be in total 6427 unique records?

We have updated the PRISMA flow diagram to reflect the updated search. The 223 additional records have been de-duplicated against existing Endnote libraries including the 6,860 unique records. 

16. Please add all reasons for exclusion of articles in the full text phase, also in Figure 1.

We have added all reasons for exclusion for Figure 1. 

17. Refer earlier in the text to Table 1, as this is providing the overview of studies. Now the first referral to the details of POCTs is to appendix S1.

Thank you. We refer to Table 1 now much earlier under Characteristics of included studies. 

18. Line 220: influenza studies were not included at all in this review, right? Why is this mentioned explicitly here?

We have deleted reference to influenza studies as this is not relevant here. 

19. Line 223-229: description of usual care is very important in order to compare the different studies, especially since this is so variable. Please add a short description of usual care to Table 1, also if ‘not specified’ in the original study.

We agree. We have now included a description of ‘usual care’ where available in Table 1 as per the description in the original text . However, we must add that in many studies ‘usual care’ was not well described and we took this to be a clinical diagnosis where no POCT was used.

20. Line 228 on antibiotic prescription of 28%: to which studies does this refer?

This is based on included studies involving children with acute fever ‘illness’ (5 studies). We have added the references and signposted readers to Figure 8. 

21. line 247: ‘principle method’, unclear terminology, be more precise. Then the immunological details are not needed in the main body of the text.

We have changed this to read ‘target analyte’ and signposted the reader to Appendix for further details. 

22. Intended use: this is also very important when comparing POCTs. Please add a column in Table 1, describing shortly what the exact decision advice was in the included studies. Which cut-off was used? What was the advised action? Also if ‘not specified’. The current categories replacement, triage etc, do not provide enough detail.

As explained above, available data did not allow us to compare studies in terms of CRP cut-offs. The description on the exact use of POCTs in included studies was either very sketchy or not available at all. However, we agree that it is important to highlight the role of the POCT in the clinical pathway (Table 1) and have now expanded on the description of these roles in the main text and more detailed in Appendix S2. 

23. Risk of bias assessment. ‘there were two cluster RCTs’; this is in contradiction with page 6 that there were 11 cluster RCTs, referring to references 23-33? At page 14 the authors refer to refs 24, 44 and 49? Then also the study of Do et al was blinded to the outcome (fig 2), so the statement that ‘one reported in two separate papers that (they?) were able to blind outcomes’ is confusing and not necessary.

There were two cluster RCTs that were able to blind outcome assessment. We have rearranged this sentence and references for clarity. 

24. Description of the outcome. See main comment 2: the way the results are presented is very confusing. First patient outcomes are reported (apparently defined as morbidity, mortality and recovery). Then under that heading other measures of impact were described as decision-making, prescribing and other testing. However, prescribing and doing other tests are also decision-making? To which decisions do the authors refer exactly? For malaria this is referral, for HIV it is initiating treatment, for fever hospital admission, for RTIs re-consultation? I suggest that the authors clearly specify in the Methods section which outcome measures of impact they have reviewed (it is currently not mentioned in the methods at all). It may be more clear to distinguish an impact on 1) diagnostics, 2) treatment and 3) disposition. This should be clear in Table 1 as well, as suggested in the previous comment 22.

Thank you for highlighting this potential ambiguity. In the Methods section, we state that we will include all quantitative clinical outcome data reporting on the impact of POCTs on clinical care and healthcare processes. We did not want to pre-empt what we were likely to find but have now outlined examples of relevant outcomes data in the Methods. We accept that there is some overlap however there is also a clear distinction between prescribing treatment and decision-making (which also includes the decision of the patient/parent to re-consult). We have now provided a more detailed description of the four distinct categories in the Method section before discussing them in detail per condition in the Results section (with the exception of patient-relevant outcomes (mortality, time to recovery) as there were so few studies. We have now clearly indicated this for the reader. 

25. Page 15 ‘in the context of safety…’ This sentence is vague, what do the authors mean hear? And why are the 3 RCTs on microscopy-positive and negative children of Table 2 not visualised in a forest plot, while their results are pooled? Other forest plots contain even fewer studies? And what is the reference standard for the effect estimate? Also after reading appendix 4 this is confusing to me. Based on the table I assume ‘microscopy results’ are the usual care? Then what is the reference standard? But in the figure of Appendix 4 the forest plot mentions POCT, when describing 3 studies on microscopy-positive children? This whole malaria treatment section needs to be revised thoroughly to improve clarity. Be consistent and specific in the terminology (POCT, usual care, rapid diagnostic tests etc)

This section is slightly complicated but is detailed in the Cochrane review of Odaga 2014 et al. examining the safety profile of POCTs versus clinical diagnosis for managing people with fever in malaria endemic settings i.e. the impact of false-negatives and positives, human error, overprescribing or clinician distrust of result. However, we agree with you that perhaps this section might detract from the overall theme and have therefore moved this entire section to Appendix S5. For clarification, we have added in a description of these to each of the four outcomes in Appendix S5. Forest plots and a detailed description are given explaining the reference standard. 

26. Page 16 – HIV section. Confusing to mention the (absent) patient outcomes reported here again, while this topic was reported on page 14. Same holds for acute RTIs at page 18.

Thanks for pointing this out. We have deleted these sentences. 

27. Page 17 – non-specific acute illness. See main comment 2. All studies evaluate the CRP-POCT, so I think it makes more sense to describe the CRP POCT here.

Please see our response above. 

28. Page 17 – decision making. ‘(…) did not show a benefit on hospital attendance or admission’. What would be the benefit: increase or reduce? Use more neutral language, since the appropriateness of the admissions cannot be judged (like ‘there was no effect on …’). This sentence reports no effect, but then the next sentence starts with ‘However…’. This implies a contradiction, please revise.

We have corrected this ambiguity by explaining the effect differences between RCTs and observational studies. 

29. Page 17 ‘We attribute the differences…’ This is for the Discussion, not results section. It does raise the question whether it makes sense at all to pool the results of such variable studies? See main comment 3.

We were careful to meta-analyse RCTs or observational studies separately. We have shifted this sentence to the Discussion section outlining the potential reasons for the effect differences as this was specific to POCT-CRP in non-specific acute illness (Interpretation of results). 

30. Page 17 – immediate antibiotic prescribing. First sentence is unclear.

We have edited this to read; “Five studies reported antibiotic prescribing.(27, 42, 45, 48, 49) using CRP-POCT. Neither RCTs (RR, 0.96; 95% CI [0.85 to 1.09], I2=0%) nor non-randomised studies (OR, 0.95; 95% CI [0.83 to 1.10], I2=0%) showed an effect on antibiotic prescribing (Fig 8).”

31. Page 19. The Danish study showed no effect on antibiotic prescribing, maybe because they had a lower baseline prescription rate? This is an important topic, please comment on this in the Discussion.

We have added this comment to the Discussion section. 

32. Page 19 – what do the authors mean with subsequent antibiotic prescriptions? What timeframe?

By subsequent antibiotic prescriptions, we mean prescriptions at re-consultation between day 3 to 5 after the initial consultation. We have clarified this in the text. 

33. Page 19 ‘there was no effect on the frequency (…), however fewer antibiotics’. To what study does the second part of the sentence refer? Another than ref 44? In its current form it suggests that this was studied in the same population.

Thanks for spotting this ambiguity. This second half of the sentence refers to Doan et al. We have re-written the sentence to clarify this. 

34. Page 20. How is the effect of strep A POCT on re-consultation a decision-making outcome? Whose decision? What was the advice of the POCT? In addition: how does the clinical protocol of making cultures in POCT negative children explain a rise in throat cultures? What was the usual care protocol then?

Here, the decision-making outcome reflects the decision of parents/carers of children to re-consult. We have clarified this in the text. In relation to throat cultures and the pre/post-implementation study by Aynaroah et al. 2009, we agree with you that this finding may not be relevant (protocol-driven). We have removed this outcome. 

35. Page 20. Diabetes. This topic is very different from all the other POCTs that focus on infectious diseases. The authors may consider removing this topic from the review, so that it has more focus. If they decide to keep it, this sections needs to be clarified: Only the trends in HbA1c levels are provided of children who used the POCT? What was the difference with usual care? Which decision was influenced by the POCT?

We agree this topic is very different from other POCTs. However, it is important to illustrate that this was the only study retrieved from our extensive search that focussed on chronic disease monitoring in children. To exclude it would detract from our aim to describe the current evidence base for POCT in children across all conditions and countries. With this in mind, we think it is important to show there is a clear evidence gap here. Readers are signposted to Appendix S7 for HbA1c levels in the usual care group. The decision influenced by the POCT relates to the clinician initiating communication with the patient between clinic visits. We have added a sentence to clarify this here and also in Table 1. 

Discussion

36. Summary of findings is very long, please summarize.

We have shortened the main findings. 

37. Page 21 bottom: to which figures to the authors refer? Are these new results?

This is an error and refers to Figures from an earlier draft. We have deleted this. 

38. Please include a section ‘interpretation of results’, in which the authors can discuss for example: how did the variability in studies influence the pooled results? What are reasons for finding a limited impact? It is mentioned a bit in the ‘comparison to the literature’ and ‘strengths and limitations’, but would be more clear if after the summary of findings the findings are shortly interpreted.

Thank you for this useful suggestion. We agree context is so important in interpreting these findings. We have included a new section “Interpretation of results”. 

39. Page 22 line ‘one explanation for this difference is that…’: logic is unclear: how can a negative test result carry a high false-positive rate? And then influence the clinicians trust? (not trusting a false-positive test would then lead to less prescriptions?)

Thank you for pointing out this ambiguous sentence. We have restructured this sentence. 

40. Page 22 ‘Another reason might be…’ this section needs a reference.

We have added a reference. 

41. Page 22 ‘Strengths and limitations’ ‘Our data were not able to evaluate if and how’ This refers to the discussion of confounders in general, that might explain the lack of evidence for impact of POCT. Not only parental concern, but also clinician’s concern etc. The section where that is described (just above strengths and limitations’ is important for the interpretation of findings, and needs more references to existing literature.

We have added moved this section to the new section “Interpretation of results” and added more references here. 

42. Page 23 first line ‘dual purpose’: this is not a dual purpose, but refers to the same decision, right? By identifying serious infections you make the prescription decision.

Not necessarily. Identifying a serious infection in ambulatory care means unscheduled hospital admission in the majority of cases. We agree that once hospitalised, a prescribing decision is also made. 

43. Page 23 Implications. The fact that there are many children in ambulatory care does not directly justify a call for more evidence for POCTs. The fact that adult studies have proven an impact does provide more justification for that call (see also comment in introduction).

Agree. We have restructured this sentence to incorporate your suggestion. 

44. Page 23 ‘POCT-negative children’. Be more specific, which POCT?

We mean malarial POCT-negative. 

45. Page 23 ‘The role of POCTs in paediatric care will differ between LMICs and HICs’. Fair point, comment on how this will be different.

We have illustrated this with two examples. These of course are not exclusive to either LMIC or HIC but in are listed in terms of priority. 

Conclusion

46. The conclusion is to broad and not supported by the rest of the manuscript. It refers to benefits in adults, that has not been clearly demonstrated in the introduction or the discussion. Moreover, it calls for cost effectiveness studies, while economic outcomes were excluded from this review.

We have rewritten the Conclusion reflecting your suggestions. 

Minor issues.

- Abstract – methods: line 54 ‘compared this…’: this suggests that the authors themselves compared the POCTs to usual care, whereas they included studies that did so. Please rephrase.

Agree. We have rephrased this sentence. 

- Introduction, line 96 ‘, this is likely’. To what does ‘this’ refer? The proportion of children? Acute illness?

This refers to the proportion of consultations. We have corrected this sentence. 

- Line 106: 20% increase in short stay. Compared to what?

We have redrafted the sentence to reflect the comparison relating to a time trend. 

- Line 111: optimise prescribing. Of antibiotics? Please add.

Added. 

- Line 206: add references of the remaining nine studies

Done.

- Line 242-244: this does not describe locations, and is already mentioned earlier, please remove.

We have removed this sentence. 

- Line 251 POCT test, remove ‘test’.

Done. 

- Table 1: there is an asterisk (*) in the row of the Nijman study that is not explained in the footnote.

Thanks for spotting this. This has been corrected. 

- After Table 1, line numbers are missing.

These are added now. 

- Figure 5 legend: please remove ‘inappropriate’, since this cannot be judged based on these data

We have removed this word. 

- Page 16 ‘the effect of HIV-POCTs (…)’. Unclear sentence. Do the authors mean that in POCT care 4 times more children initiated ARV treatment than in usual care?

We have rephrased this sentence to read “Initiating antiretroviral (ARV) therapy within 60 days in newly-diagnosed HIV children was almost 4-fold higher in those children that had an HIV-POCT compared to usual care in two studies…”.

- Page 18, acute RTIs. Add reference of the Canadian ED study.

Done. 

- Table 4 lacks a title.

We have added the following title “Table 4. Additional test investigations”. 

- Page 20 – antibiotic prescribing. Be more specific. ‘Use of the Strep A POCT did not have an impact on immediate antibiotic prescribing.’

We have redrafted this sentence to be more specific. 

- There are no references in the manuscript to Appendices 5 and 7.

We have now signposted readers to these appendices. 

Yours sincerely, 

*Oliver van Hecke (University of Oxford ) on behalf of co-authors. 

*corresponding author

---

## [Editor Report · Decision Letter 1]

19 Jun 2020

In-vitro diagnostic point-of-care tests in paediatric ambulatory care: a systematic review and meta-analysis

PONE-D-19-31785R1

Dear Dr. van Hecke,

Your manuscript has now been formally accepted for publication in PLoS One. Please see the essential details concerning the publication process below. Your efforts during the revision process are acknowledged, and I hope you are also pleased with the final result.

We appreciate being able to publish your work and look forward to seeing your paper online as soon as possible. 

Kind regards,

José Moreira, MD, MSc

Academic Editor

PLOS ONE
---

## [Editor Report · Acceptance letter]

24 Jun 2020

PONE-D-19-31785R1 

In-vitro diagnostic point-of-care tests in paediatric ambulatory care: a systematic review and meta-analysis 

Dear Dr. van Hecke:

I'm pleased to inform you that your manuscript has been deemed suitable for publication in PLOS ONE. Congratulations! Your manuscript is now with our production department. 

Kind regards, 

on behalf of

Dr. José Moreira 

Academic Editor

PLOS ONE